# HazMapper: A global open-source natural hazard mapping application in Google Earth Engine

Corey M. Scheip[1,2] and Karl W. Wegmann[1,3]

[1]Department of Marine, Earth, and Atmospheric Sciences, North Carolina State University, Raleigh, NC, 27695, USA
[2]North Carolina Geological Survey, Swannanoa NC, 28778, USA
[3]Center for Geospatial Analytics, North Carolina State University, Raleigh, NC, 27695, USA

**Correspondence:** Corey M. Scheip (cmscheip@ncsu.edu)

**Abstract.** Modern satellite networks with rapid image acquisition cycles allow for near-real-time imaging of areas impacted by natural hazards such as mass wasting, flooding, and volcanic eruptions. Publicly accessible multi-spectral datasets (e.g. Landsat, Sentinel-2) are particularly helpful in analyzing the spatial extent of disturbances, however, the datasets are large and require intensive processing on high-powered computers by trained analysts. HazMapper is an open-access hazard mapping application developed in Google Earth Engine that allows users to derive map and GIS-based products from Sentinel or Landsat datasets without the time- and cost-intensive resources required for traditional analysis. The first iteration of HazMapper relies on a vegetation-based metric, relative difference of normalized difference vegetation index ($rdNDVI$) to identify areas on the landscape where vegetation was removed following a natural disaster. Because of the vegetation based metric, the tool is typically not suitable for use in desert or polar regions. HazMapper is not a semi-automated routine but makes rapid and repeatable analysis feasible for both recent and historical natural disasters. Case studies are included for the identification of landslides and debris flows, wildfires, pyroclastic flows, and lava flow inundation. HazMapper is intended for use by both scientists and non-scientists, such as emergency managers and public safety decision-makers.

## 1 Introduction

Natural disasters such as landslides, wildfires, and volcanic eruptions are a primary mechanism of landscape change (Korup et al., 2010; Santi et al., 2013) while simultaneously causing fatalities into the 21$^{st}$ century (Froude and Petley, 2018; Petley, 2012; Auker et al., 2013; Holzer and Savage, 2013; Ashley and Ashley, 2008). As such, both Earth scientists and emergency managers have a keen interest in understanding natural disaster occurrences and their spatial extent. Ongoing work to increase predictive capabilities for natural hazard events (Goetz et al., 2015; Guzzetti et al., 2006) rely on the robust characterization (e.g. number or spatial distribution of landslides) of modern-day events (Xu et al., 2016; Gallen et al., 2017).

Readily observable field evidence of only the largest or most recent natural disasters typically persist in temperate environments due to the constant regrowth cycle of vegetation. Locating this field evidence and determining historical patterns of

natural disasters is a primary objective for many agencies and communities (Paton and Johnston, 2001; Wegmann, 2006; Brand et al., 2019). However, field work is often inefficient and expensive compared to remote sensing methods and it does a poorer job at temporally constraining natural disasters. For example, the timing and extent of a series of landsliding events over many years will be difficult and expensive to determine with field methods alone. In addition, field-verified inventories of spatially extensive events may take many months to years before their completion. In this case, remote sensing methods may be able to temporally discern the cycle of events (Gold et al., 2004) and reduce the lag time between event occurrence and inventory development.

The advent of rapid-repeat cycle satellite datasets (e.g. multi-spectral, thermal) revolutionized the field of remote sensing and our ability to observe landscape changes. These data have been utilized since the 1970's (e.g. Landsat) to observe, monitor, and track landscape change (Lauer et al., 1997). In 2008, NASA began offering Landsat datasets for free to the general public via the internet (Woodcock et al., 2008) and by now, the entire archive is available online. Subsequent satellite networks and payloads (e.g. MODIS, Sentinel) have improved capabilities and increasingly advanced satellite networks continue to be developed (e.g. Hoffman et al., 2016; Langhorst et al., 2019). Some traditional uses for satellite-observations include identifying landslides and debris flows (e.g. Tillery and Rengers, 2019), wildfire (e.g. Miller and Thode, 2007; Amos et al., 2019), volcanic monitoring (e.g. Cando-Jácome and Martínez-Graña, 2019), deforestation (e.g. Hansen et al., 2013; Green and Sussman, 1990; lan), urban change and development (e.g. Masek et al., 2000; Schneider, 2012), and ecological monitoring and change detection (e.g. Zhou et al., 2001; Meentemeyer et al., 2004), amongst others.

Analysis of remote-sensing data has traditionally been performed by trained analysts on high-powered computers, which can create a resource-barrier for fiscally strained communities or those without advanced training. Combining the open-access nature of these datasets with modern computational power available via cloud-computing is a recent development that has powerful implications for natural disaster monitoring (Kirschbaum et al., 2019). Google Earth Engine is a remote sensing data analysis platform designed to take advantage of Google's infrastructure for data storage, access, processing, and visualization (Gorelick et al., 2017).

HazMapper (Hazard Mapper) is an open-access application developed in Google Earth Engine for the rapid characterization of natural disasters tailored to both the scientific and emergency management communities (Figure 1). HazMapper is useful for monitoring landscape change that results in the removal or recovery of terrestrial vegetation associated with a natural disaster or human activities. The platform is not currently suitable for use in non-vegetated environments (e.g. polar, high altitude, or desert regions).

While the underlying mathematics are not novel, HazMapper democratizes multi-spectral satellite data processing for the evaluation of natural hazards by leveraging the accessibility and computational power of Google Earth Engine. Select case studies are discussed here and include both rainfall and seismically-triggered mass wasting, wildfire, pyroclastic flow, and landscape burial by lava flows resulting from a fissure eruption. In the aftermath of large natural disasters, the level of emergency response can vary based on available resources for the region or country. For example, response efforts for a large landslide disaster in the United States, (e.g. 2014 Oso landslide, WA or 2019 Montecito, CA debris flows) can garner the attention and resources of local, state, and federal government agencies (Scholl and Carnes, 2017). In less affluent regions,

however, response efforts may be less intensive and timely, risking increased loss of life. An overarching goal of HazMapper is to bring modern, rapid scientific analysis and computing power to regions with less adequate resources. HazMapper is publicly accessible at https://hazmapper.org/.

## 2   Design Principles

Traditional GIS analysis consisting of downloading satellite images and processing on local machines has several limitations in the modern computing environment. These limitations are exacerbated when attempting to perform spatially expansive analysis, such as identifying far-field effects of a natural disaster (e.g. analyzing hundreds of kilometers around an earthquake epicenter for coseismic mass wasting). Further, the potential need to observe many pre- or post-event images due to challenges such as cloud cover or atmospheric aerosols can hinder this workflow by requiring the analyst to repeat the data download-process routine. This fixed-extent processing routine inhibits the analyst from exploring impacts without stopping analysis to download and process additional data.

In the context of these limitations, HazMapper was developed in Google Earth Engine because the platform provides some key advantages compared to traditional download-process routines. For example, the platform's architecture initiates geospatial processing updates as the user navigates the map. Because of this feature, HazMapper can be used to quickly evaluate spatially expansive hazards by panning or zooming without downloading any data. The source code for HazMapper initiates data processing on remote servers without requiring any specialized or licensed software and can be performed on any internet-connected device. Typical processing times on HazMapper are less than 1-2 minutes, allowing users to adjust variables and visualization parameters to rapidly assess the potential impact of natural hazards at a specific location or across a region of interest, and to test hypotheses about the spatial and temporal extent of a natural hazard event. Google Earth Engine allows users to create public-facing applications (such as HazMapper), further increasing the accessibility of processing routines to specialists in the field as well as the scientifically curious public.

HazMapper is designed around user-input variables in a graphical user interface (GUI). The user is able to modify the following variables: dataset selection (Landsat 7, Landsat 8, Sentinel-2), event date, pre-event time window, post-event time window, maximum cloud cover for analysis, and slope thresholds (Table 1). These variables can be updated throughout the use of the tool to aid in identifying hazard-stricken locations. The basemap options are Google's global terrain or satellite imagery. The resulting data layers are displayed in a layers pane, available for toggle and transparency adjustment.

Traditional remote sensing-based landslide mapping is performed by analysts observing change in aerial photographs (e.g. Malamud et al., 2004; Wegmann, 2006; Abancó et al., 2020). This method relies on single pre- and post-event scenes and is hampered when unfavorable atmospheric conditions exist. In the immediate aftermath of natural hazards that are initiated atmospherically (e.g. rainfall-triggered mass wasting) or volcanically (e.g. eruptions), it is not uncommon for imagery to be obscured by clouds or ash plumes, respectively. In cases of tropical or subtropical cyclones, cloud cover may persist for days to weeks following a disaster. Additionally, tropical regions of elevated rainfall and topography, and thus typical of mass wasting or flooding hazards, may experience cloud-cover for a substantial portion of the annual climate cycle.

To overcome potentially opaque atmospheric conditions that might be caused by aerosols including water vapor (clouds), smoke, mineral matter (dust), or anthropogenic pollution (smog), HazMapper capitalizes on a technique within Google Earth Engine to generate and perform calculations on a greenest-pixel composite (Figure 2). The compositing method utilizes data from many images, reducing artifacts present from clouds and other aerosol particles in a given single image. The greenest pixel composite is a single composite or tiled image generated from all images within the user-defined pre- and post-event window that records the pixel with the highest normalized difference vegetation index (NDVI) result, or the "greenest" pixel (Eq. 1).

$$NDVI = \left( \frac{NIR - VIR}{NIR + VIR} \right) \tag{1}$$

where NIR is the near-infrared response and VIR is the visible infrared response.

HazMapper relies on a relative difference in NDVI technique ($rdNDVI$, Eq. 2). Instead of differencing true color composites (i.e. red-green-blue bands), HazMapper exploits changes in surface vegetation by developing and differencing a NDVI-band from the greenest-pixel composite images,

$$rdNDVI = \left( \frac{NDVI_{post} - NDVI_{pre}}{\sqrt{NDVI_{pre} + NDVI_{post}}} \right) \times 100 \tag{2}$$

where NDVI$_{pre}$ and NDVI$_{post}$ are the NDVI images of the pre- and post-event greenest pixel composites, respectively.

The results of the processing routine indicate a normalized percentage of NDVI gained or lost. The normalization parameter includes $NDVI_{pre}$ to account for pixels that had a low NDVI value before the event (e.g. existing mass wasting scars, urban areas). The metric follows Norman and Christie (2020), who propose this method for addressing fractional pixels, where the NDVI signal response of vegetation is not consistent across a single pixel (e.g. both forest and grasses), and the non-linear responsiveness of NDVI. Results may exceed +/- 100%. $rdNDVI$ results of < -100% is possible due to the ability of VIR to increase to greater than the NIR value, causing a polarity change of NDVI. Results illustrate areas of the landscape that have either gained (increase in NDVI pixel-values) or lost (decrease in NDVI pixel-value) vegetation across the event as constrained by the pre-and-post event window date ranges. For visualization purposes, HazMapper applies a color-scale within the domain of -50% to +50%, simplifying the analysis to highlight areas with vegetative loss or gain. Additionally, an inspector tool allows a user to click anywhere within the map domain, upon which a latitude-longitude coordinate pair and the $rdNDVI$ pixel value will be returned interactively to the user.

The three resulting data layers (greenest pixel composite from pre- and post-event, and $rdNDVI$) and a shuttle radar topography mission (SRTM) derived 30-m resolution hillshade layer (for areas between 60°N and 60°S latitude) are added to the standard Google Earth Engine layer pane. These layers can be toggled on/off and their transparency modified with a slider to help with visualization.

Digitization of areas of interest from the map domain, for example, debris flow initiation sites, landslide extents, or potential wildfire burn areas, can be recorded using Google's default mapping tools. Points, lines, and polygons may be digitized in one

or multiple layers. During download from HazMapper, these digitized geometries can be saved as a keyhole-markup language (KML) file for viewing in Google Earth or sharing amongst an emergency response team or between research colleagues.

HazMapper includes an example panel in the lower left of the tool, pointing the user to five real-world natural hazard case studies (Figure 1). The panel is intended to serve as a learning platform for new users to work with curated examples to explore these events.

Additional sharing of HazMapper results is made available via the use of variable-tags within the URL. During usage of the tool, URL tags for the required event parameters are updated. Sharing of the updated link with a colleague or research partner allows that person to open HazMapper to the same viewport and updates the map function to the same event parameters. For the case studies discussed here, we have utilized the North Carolina State University Go Links URL service. For example, https://go.ncsu.edu/hazmapper-kenya directs the user to the curated example of rainfall-triggered debris flows located in west Pokot County, Kenya, in November 2019 (see Section 4.1.1). All case studies discussed in this paper are available pre-loaded from the HazMapper launch screen, under the "Show Examples" tab, or as direct URL links (see *Code and Data Availability*.

Data download is an important component of HazMapper. This function allows for further analysis of processing results, including incorporation into emergency operation mapping platforms and advanced scientific analysis or visualization. The user can download the 1) $rdNDVI$ image, 2) pre-event and 3) post-event greenest-pixel composite images, 4) elevation and hillshade images derived from the global 30-m SRTM dataset and/or 5) any user-digitized geometries delineating points or areas of interest. Raster data layers are distributed as geographic tagged image file format (GeoTIFF) and digitized geometries as KML.

## 3 Results – Case Study Examples

The five curated examples included in the example panel (lower left of tool) are discussed herein. The intent of these case studies is not to provide an exhaustive analysis of the events, but to showcase various applications of a $rdNDVI$ cloud-computing method.

### 3.1 Mass Wasting

Landslide events are a primary contributor to topographic erosion and landscape evolution (Korup et al., 2010), make available significant rock-bound and organic (soil and above ground biomass) carbon for global biogeochemical cycling (Hilton et al., 2008), and caused at least 55,997 non-seismic landslide fatalities between 2004-2016 and billions (USD) in global losses and damaged infrastructure costs (Froude and Petley, 2018; Emberson et al., 2020; Kirschbaum et al., 2015; Petley, 2012). In the United States alone, annual losses to mass wasting events exceed $3 billion (Spiker and Gori, 2003; Burns, 2007).

Significant amounts of research has been performed on remotely detecting mass wasting events such as debris flows, debris slides, or rock slides (Kirschbaum et al., 2019; Amatya et al., 2019; Mondini et al., 2011; Lu et al., 2019; Huang et al., 2020; Tsai et al., 2010; Yang et al., 2013). Pixel-based or object-oriented analysis (OOA) methods rely on characterizing change to the Earth surface via multi-spectral satellite imagery and correlating these changes to mass wasting events (e.g. Lu et al., 2019).

Recognizing that in forested areas, landslides denude the landscape of vegetation, NDVI change detection methods have been used for identifying landslides in many mid-latitude regions (Huang et al., 2020; Tsai et al., 2010; Mondini et al., 2011; Lu et al., 2019; Yang et al., 2013).

### 3.1.1 Rainfall-triggered debris flows, West Pokot County, Kenya, 23 November 2019

In mid-late November 2019, regions in eastern Africa experienced many days of intense rainfall. West Pokot County, Kenya, located in the rugged terrain of the east African rift valley in western Kenya (Figure 3), received greater than 400 mm of rainfall within the period 23-25 November Huffman et al. (2014). This rainfall event triggered failures of steep, soil mantled hillslopes along the western rim of the east Africa Rift Valley (Elgeyo Escarpment), initiating fatal debris flows that destroyed homes, agricultural fields, and infrastructure.

As early as December 4th, 2019 (11 days following the mass wasting event), suitable Sentinel-2 datasets were available and HazMapper was utilized to locate the debris flows (Figure 3; see Supplement). In addition to the location of debris flows, $rdNDVI$ also captured riparian vegetation loss and sedimentation along the banks of lower-gradient rivers as they drain the mountainous terrain where the mass wasting occurred. Agricultural harvesting and planting activities are also apparent in the $rdNDVI$ results, evident by their position on low-relief terrain outside of the drainage channels (Figure 3D).

### 3.1.2 Seismically-triggered mass wasting, Southern Highlands, Papua New Guinea, $M_w$ 7.5, 25 February 2018

On 25 February, 2018, a $M_w$ 7.5 earthquake struck the Southern Highlands of Papua New Guinea (PNG) along the Papuan Fold and Thrust Belt (Wang et al., 2019), triggering thousands of mass wasting events, damming the Tagari River, and impacting numerous communities across the region. Over a span of 2 months, 5 aftershocks of $M_w$ >6 struck the same region (Wang et al., 2019). Nearly three years after the event, a published mass wasting inventory is not available. Fatalities from coseismic mass wasting can be up to an order of magnitude greater than fatalities resulting from the earthquake itself (Budimir et al., 2014). The 2018 Papua New Guinea earthquake and associated mass wasting resulted in at least 160 fatalities (Wang et al., 2019), but the individual contributions (e.g. building collapse, burial by hillslope mass movements, etc) are not well understood on account of this event occurring in a rural and remote part of the country.

Seismic shaking is a primary triggering mechanism for mass movement mobilization on steep mountain terrain. Coseismic mass wasting, therefore, strongly influences erosional budgets of mountain belts (Hovius et al., 1997; Keefer, 1994; Korup et al., 2010; Hilton et al., 2008). Keefer (2002) has demonstrated an empirical relationship based upon a global dataset between the moment magnitude of a mainshock and the maximum distance from the epicenter that seismically-induced landslides are likely to be observed for the entire earthquake sequence (including aftershocks). For this $M_w$ 7.5 earthquake, the corresponding predicted maximum distance is approximately 300 km. HazMapper was utilized to rapidly assess regions within several tens of kilometers from the epicenter and hundreds of slides and flows were located (Figure 4). Additional mass wasting was noted when expanding the analysis window to the predicted 300 km maximum distance based on the earthquake magnitude. Furthermore, we noted possible coseismic slides and flows as far as several hundred km west of the epicenter in the Maoke Mountains of Indonesia. Mass wasting is common in the region and these events could have unique triggers (e.g. rainfall

triggered). However, restricting pre- and post-event time windows to as little as 2 months bracketing the $M_w$ 7.5 mainshock demonstrates consistent timing with the 25 February, 2018 earthquake.

Due to difficulties in ascertaining high-temporal-resolution sequences of mass wasting events following seismic shaking, it is typically difficult to determine if particular events were triggered by just the mainshock or also by aftershocks. Thus, research to date has focused on earthquake sequences, inclusive of all associated shaking (e.g. Keefer, 2002). HazMapper allows researchers to temporally constrain landscape change and in certain circumstances, may be useful for understanding hillslope failure sequences for large earthquakes that may include landslide inducing aftershocks. Future research should consider utilizing these time-stamped change detection images to understand the progression of failures during an earthquake sequence.

While identifying vegetation loss for locating geohazards is a key characteristic of a mass wasting event response, identifying subsequent vegetation recovery can serve as a proxy for the reduction of associated hazard (Shen et al., 2020). Simple modifications to event parameters in HazMapper, for example by changing the "event date" to a time after the occurrence of the disturbance event, can aid in observing vegetation recovery in landslide scars, suggesting establishment and growth of early successional species like grasses and shrubs (Figure 4-D). These stabilizing root masses buttress further soil loss and erosion, and, thus, decrease the associated downslope sediment transport from the zone of mass wasting.

## 3.2 Wildfire

Wildfire experts have been utilizing multi-temporal, multi-spectral imagery to evaluate burn extents following wildfires since at least the launch of the Landsat thematic mapper program in 1984 (Keeley, 2009; Miller and Thode, 2007; Cocke et al., 2005). Multi-spectral indices such as the popular Normalized Burn Ratio (NBR) and its derivatives (e.g. difference, relative difference) are widely employed to assess ecosystem impacts following a wildfire (Miller and Thode, 2007; Cocke et al., 2005). Discussion and debate about the most appropriate multi-spectral index to utilize for understanding fire impacts in wildland fire science are ongoing (Keeley, 2009; Miller and Thode, 2007; Escuin et al., 2008; Amos et al., 2019). The $rdNDVI$ technique (Equation 2) utilized in HazMapper is one such index (Norman and Christie, 2020).

### 3.2.1 Chimney Tops 2 Fire, Tennessee, USA, November 2016

In the autumn of 2016, the southern Appalachian Mountains experienced drought conditions that contributed to dozens of wildfires that totaled some 75,000 acres (Andersen and Sugg, 2019). Originating within the Great Smoky Mountains National Park (GRSM), the Chimney Tops 2 fire was first discovered on 23 November, 2016 (National Park Service, 2017; Jiménez et al., 2018). The fire initially ignited on top of the north spire of Chimney Tops inside GRSM (Guthrie et al., 2017). Unexpected wind conditions facilitated the rapid expansion of the fire perimeter and fires were noted inside the city limits of Gatlinburg, TN, 10 km from the ignition point, by 28 November (National Park Service, 2017; Guthrie et al., 2017). The Chimney Tops 2 fire burned some 17,000 acres before eventually merging with other eastern Tennessee wildfires (Guthrie et al., 2017). Impacts from the fire included 14 fatalities, 14,000 evacuations, over 2,500 structures lost, an estimated $2 billion in damages, and was the largest wildland fire in recorded history in the park (National Park Service, 2017; Guthrie et al., 2017).

HazMapper was utilized to observe vegetation loss, and by proxy, the severity and burn extent (Figure 5). In addition to the simple fire perimeter, the HazMapper method also illustrates burn severity by way of $rdNDVI$, highlighting that the most severe burn, as indicated by greater percent $rdNDVI$ decreases, occurred along ridges and upper elevations, consistent with typical wildland fire behavior (Teie, 2018).

Following on the identification of vegetation loss, subsequent post-fire vegetative re-greening of the landscape is depicted as $rdNDVI$ increases. Figures 5-B and 5-C illustrate recovery between the first through second growing seasons following the fire (2017 to 2018) and from the first through the third growing seasons (2017 to 2019), respectively. Forest recovery monitoring via remote sensing data is not a novel approach (Chen et al., 2014; Cuevas-González et al., 2009), however, the rapidity of observing the recovery via an open-access remote-processing and cloud-based platform is, to our knowledge, novel.

### 3.3 Volcanic Eruptions

Between 1600 to 2010 CE, 533 volcanic events have resulted in at least 278,880 fatalities (Auker et al., 2013). The number of fatalities each year attributable to volcanic events is increasing with time (Auker et al., 2013), suggesting that as a species, we are not overcoming the danger associated with volcanic hazards. As such, volcanologists have been using remote-sensing tools, particularly multi-spectral satellite data, as early as the mid-1980's to monitor volcanic heat signatures as precursors to eruptive activity (Rothery et al., 1988). The moderate resolution imaging spectrometer (MODIS) multi-spectral sensor is commonly used to monitor thermal characteristics and to detect volcanic eruptions (Wright et al., 2002), however, it's variable 250 to 1000 meter pixel size inhibits the use of the platform for adequately identifying many downslope hazards associated with eruptions. Downslope hazards may include lava flows, ballistic projectiles, pyroclastic flows, and lahars (Blong, 1984). Following eruptions, HazMapper's use of 30-meter Landsat or 10-meter Sentinel-2 data is well suited to identify the spatial extent of these hazards, which may be only meters to tens of meters wide (e.g. a lahar track) and may travel many kilometers from the volcano.

### 3.3.1 Pyroclastic Flows — Volcan de Fuego, Antigua, Guatemala, 3 June 2018

Volcan de Fuego is a subduction zone stratovolcano located in southwestern Guatemala, near the city of Antigua. Since 1524, Fuego has produced 51 eruptions with a volcanic explosivity index $\geq 2$ (Global Volcanism Program, 2013). The volcano is renowned for its consistent low-intensity Strombolian eruptions punctuated by larger, more violent sub-Plinian eruption cycles (Naismith et al., 2019). The most recent eruptive cycle of Fuego, ongoing since 2015, consists of an increase in paroxysmal eruptions and resulting downslope hazards (Naismith et al., 2019). The 3-5 June 2018 sub-Plinian eruption generated pyroclastic flows in excess of 11 km in length, resulted in hundreds of fatalities, and destroyed the rural community of San Miguel Los Lotes (Pardini et al., 2019; Naismith et al., 2019).

HazMapper was utilized to observe landscape change following the 3-5 June 2018 Volcan de Fuego eruption (Figure 6). Summit effects of the eruption are observed via HazMapper, as well as pyroclastic flows down the west, south, and east flanks of the stratovolcano for up to 11km. Due to Fuego's consistent eruptive activity and loose, steep volcanic sediments, vegetation is generally sparse near the summit (Figure 6-B). However, as indicated by the $rdNDVI$ results, loss of the limited-vegetation

was evident on 30-meter Landsat data. Pyroclastic flows from the 3-5 June, 2018 eruption are evident in the main valleys draining away from the volcano. Riparian vegetation loss and sedimentation was noted for >60 km to the south-southwest from the volcano, and in one case, this impact is observed as far as the confluence with the Pacific Ocean (Figure 6-A).

**3.3.2   Effusive lava flows — Lower East Rift Zone (LERZ), Kīlauea Volcano, Hawaii, USA, May-September 2018**

Kīlauea is a basaltic shield volcano built from lavas derived from deep mantle driven processes. The magma feeding the volcano is distributed through a network of shallow rift structures and was pooled in a lava lake at its summit until commencement of the 2018 eruptive sequence. Eruptive characteristics have varied through time including a combination of periods of summit and/or rift eruptions, and caldera collapse, in-fill, and overflow (Holcomb, 1987). The most recent 2018 caldera collapse-rift eruption

sequence was well captured by a dense array of scientific instrumentation and social networking, adding significant information to our present understanding of the Kīlauea complex (Neal et al., 2019). The 2018 event culminated in the inundation of 35.5 km$^2$ of Hawaii's Big Island and the destruction of hundreds of homes. Fortunately, there are no known fatalities from the event, likely due to the slow moving nature of the eruption and the significant resources applied during the disaster management response efforts.

HazMapper was utilized to observe surface changes within the Lower East Rift Zone (LERZ) following the cessation of the rift flank eruption sequence (Figure 7). Utilizing 30-meter Landsat data, the observed vegetation loss extending east and southeast from the LERZ approximates the published flow field from the 2018 eruption (Hawaiian Volcano Observatory staff, 2018). Efforts to utilize HazMapper to monitor the advance of the lava flows were hindered due to persistent cloud cover and volcanic gas emissions during the eruption. Additionally, the east-southeast extents of the lava flows generated additional

landmass off of the coast of Hawaii, but with no vegetation to lose, this landscape change was not detected using HazMapper.

The 2018 Kīlauea eruption response benefited from significant resource application by way of the existing Hawaii Volcano Observatory and the associated resources of the U.S. federal and Hawaii state governments and associated scientific and resource protection agencies. This example, therefore, is highlighted to perform a first-order comparison of the kind of results available with HazMapper, a free and open-access toolset to an on-the-ground effort with significant financial, personnel,

equipment, and computing resources and attention (Figure 7). For eruptions with less global attention or in more remote regions, remote sensing results like those available with HazMapper alone may approximate lava flow inundation extents, guiding future response efforts or scientific research around the event. Furthermore, the utilization of a consistent analysis platform between many eruptions may aid in volcanic research globally.

## 4   Discussion

**4.1   Limitations**

In the era of big data and cloud computing, a utility like HazMapper increases the pace at which researchers can evaluate global natural hazards. However, as with any analysis platform, limitations exist. The most important limitation of HazMapper,

at present, is that the platform is neither an automated nor semi-automated routine. The platform does not predict where landslides have occurred, or what areas were burned during a wildfire, for example. Instead, the platform computes a very

simple vegetation based metric, which in many cases can be indicative of a natural disaster. As such, no accuracy assessments or omission-commission errors are included in the case studies presented here. HazMapper does, however, provide a solid platform for the development and use of future semi-automation routines for various natural disasters. Augmentation and testing of user-driven semi-automation techniques utilizing machine learning in HazMapper are in the developmental stage and will be discussed in a future manuscript.

The underlying data sets in HazMapper allow users to constrain the timing of an event to $\geq 5$ days (Sentinel-2) or $\geq 16$ days (Landsat). Recognizing that during natural disasters, community impacts can undergo many developments in this time frame, differences in pre- and post-event images may require some interpretation and cannot provide a day-by-day account of ground conditions. This must be considered in the context of alternative approaches, however. Traditional remote sensing studies often rely on single pre- and post-event images. In these cases, it is an assumption of the analyst that the impacts occurred during

the event being studied. For example, if a landslide is present in a post-event single image, but not a pre-event single image, it is often assumed that the landslide occurred during the particular event under consideration (e.g. Huang et al., 2020; Lu et al., 2019).

The timing of suitable data sets for an initial evaluation of disaster impacts will depend on the timing of a event relative to acquisition schedules of the Sentinel-2 or Landsat platforms, atmospheric conditions (e.g. cloud cover during or following a

rainfall-triggered mass wasting event, or smoke from long-burning wildfires), and seasonal considerations. Acquisition schedules are publicly available for both platforms and can be used in conjunction with HazMapper to help responders understand when suitable data sets may become available. Even if the imagery is not clear (e.g. cloudy, obscured by aerosols), performing the analysis generally takes seconds and, because of the URL-update feature, provides a URL that can be saved offline and re-executed when new data are posted. Events that occur in the winter in temperate environments can be difficult to discern

because there is minimal healthy green vegetation to lose during a disaster. In these cases, observing change in the pre- and post-event color composite images may be helpful, however, analysis following the spring green-up of the landscape often will provide a more complete understanding of impacts.

HazMapper only accesses data sets publicly hosted within the Google Earth Engine Catalog. While many researchers have the funding to pursue the use of data sets acquired on a near-daily basis (e.g. from Planet Labs, Inc.), HazMapper currently

does not have a mechanism for ingesting these data.

Two final limitations relate to the spatial extent of processing and data downloads. Even with the large amounts of computational power available with the platform, continental scale analysis is typically not feasible. Long wait times for data or its failure to load in HazMapper commonly results when the user is trying to evaluate too large of an area. Zooming in will initiate reprocessing and data will usually begin to load. Similarly, current external user downloads (e.g. users of the application) are

limited to 32 megabytes by Google. This is approximately 400 square kilometers with Sentinel-2 (10-meter pixel size) data. HazMapper utilizes the entire view extent within the user's browser window when the download routine is initialized. If the

area is larger than 400 square kilometers, the download window may not appear. If this occurs, the user should zoom in to a larger scale.

With all of these limitations, it is important to recognize that Google is regularly making improvements and modifications to Earth Engine. We intend to monitor these activities and update the HazMapper application as needed to mesh with future changes to Earth Engine.

## 4.2   Use of the Event Parameters

The event parameters window is designed to allow flexibility in analysis for various natural disaster types, geographies, and objectives. The combination of the GUI layout and URL-updating is designed to promote rapid analysis of events with different parameter combinations to best understand the natural disaster. Event parameters are defined in Table 1, and details regarding their usage are provided here:

– **Dataset.** This parameter allows the user to determine which data set (Landsat-7, Landsat-8, or Sentinel-2) to use for analysis. The coarser resolution of the Landsat sensors (30-meters) compared to the Sentinel sensors (10-meters) is faster to process and is suitable for analysis of larger regions (e.g. Section 3.3.2). The higher resolution Sentinel sensors are more applicable for identifying smaller features, such as narrow debris flow tracks). Landsat-7 is included in HazMapper to allow analysis of events that occurred as early as 1999. Following the May, 2003 scan line corrector (SLC) mirror failure, individual Landsat-7 images exhibit striping, however, the greenest pixel compositing methods of HazMapper can be leveraged to mitigate these artifacts. By extending window lengths, these gaps are filled with data from subsequent overpasses. As an example, Figure 8 evaluates erosion, vegetation removal, and sedimentation along a coastline of Indonesia following the 2004 Sumatra-Andaman seismogenic tsunami. A short window length (1 month pre- and post-event) analysis is obscured by striping. By extending the window lengths, these artifacts are reduced.

– **Event Date.** The event date is intended for demarcating the pre- and post-event periods for which to perform the $rdNDVI$ calculation. Generally this is a known value from the literature or news reports. However, by combining this field with imagery acquisition schedules, this parameter can also be used to determine the date of a natural disaster. Date estimates can only be made within the margin of error that is approximated by the sensor's acquisition, (e.g. within 5 days if using Sentinel-2, or within 16 days if using Landsat).

– **Window Lengths.** Window lengths are measured in months and can be altered to capture more or less data before or after an event. This is useful when balancing the need for cloud-free imagery with a level of confidence in the timing of impacts from a natural disaster. For example, a small window length will allow the user to be confident in the timing of the natural disaster impacts. For vegetation loss studies, this confidence is driven by the pre-event window length. This is because of the greenest pixel compositing technique, which always identifies the maximum-NDVI conditions in the given analysis window. For example, the pre-event window length in the Kenya debris flow example (Section 3.1.1, Figure 3) temporally constrain the debris flows to have occurred within the 2-month period preceding the event date of 23 November 2019 (+/- 5 days, consistent with the acquisition rate of Sentinel-2). Combining this result with news

reports, interviews, or otherwise local knowledge of the region can increase confidence that the major debris flow swarm imaged in HazMapper was a result of the 23 November 2019 storms.

- **Maximum Cloud Cover.** The intent of this parameter is to increase processing speed by omitting any images with cloud cover greater than the parameter value. For example, if the user is confident there will be images with very low cloud cover, a value such as 30% maximum cloud cover will only perform the cloud filtering and $NDVI$ calculations on those images with cloud cover parameters less than 30%. This will also yield the cleanest results, because even the cloud filters utilized in Earth Engine can result in some peripheral noise in the composite or $NDVI$ images.

- **Slope Threshold.** This minimum slope value for analysis is used to mask out areas of low slope during the visualization process. This can be very useful to remove water bodies (e.g. lakes, oceans) which are often noisy and can distract from $rdNDVI$ targets. Similarly, this can be used to mask out low slope areas, like wide valley bottoms, while processing an area for mass wasting events.

## 4.3 Further Applications

HazMapper may be useful in many other types of studies than those described here. For example, vegetative can be a proxy for decreasing hazard following large mass wasting events (Shen et al., 2020; Yunus et al., 2020). By altering the parameters in HazMapper, this regrowth can be tracked and quantified, e.g. Figure 4D. Additional applications include characterizing sedimentation patterns, agricultural or logging operations (including tracking illegal operations), monitoring controlled burns and the growth of early successional species, or biological blight monitoring.

## 4.4 Future Work

There are several anticipated development opportunities for HazMapper. Principal amongst these is the development of hazard-specific platforms to provide more focused analysis for various hazard types. For example, a mass wasting platform could incorporate pixel segmentation, consideration of developed areas, and slope thresholds. Further research into the application of HazMapper in arid or snow-covered environments, including the consideration of snow-related indices (i.e. normalized difference snow index, NDSI), is on-going, and may be helpful for detecting mass wasting events in non-vegetated, high-latitude, or high-elevation regions. A wildfire platform can be expanded to include burn-specific indices such NBR. And for all platforms, various change detection methods (e.g. short-wave infrared differencing, amongst others) should continue to be evaluated. Radar data brings exciting opportunities, and as radar processing routines become available within GEE, they should be leveraged (e.g. (Handwerger et al., 2020)).

Future development of HazMapper will leverage new data sets as they become available. The initial release includes options to analyze Sentinel-2, Landsat 8, and Landsat 7 data sets. HazMapper's underlying source code is designed to easily incorporate multi-spectral imagery data from forthcoming missions, such as Landsat 9 that is anticipated to launch in December 2020 (McCorkel et al., 2018).

## 5 Conclusions

HazMapper is a free and open-access application developed in Google Earth Engine. It is primarily tailored to observing vegetated-landscape change as a proxy for natural hazard impacts. The approach is novel, leveraging the power of Google Earth Engine to democratize change detection from multi-spectral satellite imagery in a user interface designed for researchers, emergency responders, and the scientific-curious public. HazMapper does not require users to download any datasets, posses a background in data analysis, software development, or coding, or have access to specialized software other than an internet-browser. And because processing occurs remotely, low-powered computers (e.g. Google Chromebooks), tablet computers, and even smart phones are suitable for use with HazMapper; although the small screen size of such devices is a limitation.

In version 1.0 of HazMapper, a single toolset is released for the observation of surface vegetation loss by way of a relative difference in NDVI values, suggesting the extents of hillslope and channelized mass wasting, wildfires, pyroclastic flows, and lava inundation areas. The is not a semi-automated method and HazMapper does not predict areas of natural disasters. Instead, the platform currently makes $rdNDVI$ calculations accessible and performs them rapidly, increasing the pace at which researchers can evaluate events. HazMapper is an open-source project and community contributions are welcomed. Supporting JavaScript source code for HazMapper is available at https://doi.org/10.5281/zenodo.4103348.

*Code and data availability.* Source code is available under a research only license at https://doi.org/10.5281/zenodo.4103348. Examples discussed in this article or presented in figures can be accessed from the following URLs.

- Rainfall-triggered mass wasting, West Pokot County, Kenya, 23 November 2019: https://go.ncsu.edu/hazmapper-kenya
- Seismically-triggered mass wasting, Southern Highlands, Papua New Guinea, M 7.5, 25 February 2018: https://go.ncsu.edu/hazmapper-png
- Chimney Tops 2 Wildfire, Tennessee, USA, November 2016: https://go.ncsu.edu/hazmapper-chimneytops
- Pyroclastic flows, Volcan de Fuego, Antigua, Guatemala, 3 June 2018: https://go.ncsu.edu/hazmapper-fuego
- Lower East Rift Zone eruption, Kīlauea Volcano, Hawaii, USA, May-September 2018: https://go.ncsu.edu/hazmapper-lerz
- Tsunami inundation, Indonesia, 26 December 2004: https://go.ncsu.edu/hazmapper-indonesia-tsunami

*Author contributions.* CMS and KWW contributed to the design and execution of building HazMapper, including the source code and this manuscript. Both authors gave final approval of this manuscript.

*Competing interests.* The authors declare that no competing interests are present.

*Acknowledgements.* We gratefully acknowledge Steve P. Norman of the U.S. Forest Service Southern Research Station in Asheville, NC for helpful early conversations regarding the mathematics behind the $rdNDVI$ techniques. Additional thanks are owed to Dr. Jesse Hill of the

North Carolina Geological Survey for helpful discussions regarding the use of the $rdNDVI$ technique in observing debris flows in western North Carolina and for his input on the HazMapper user interface. The comments and suggestions of two anonymous reviewers assisted in the improvement of the initial submission.

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

**Table 1.** HazMapper input variables, definitions, and examples.

| Input Variables | Definition | Example |
|---|---|---|
| Dataset | Dataset to use for analysis. Currently Landsat 7, Landsat 8, or Sentinel-2 | Sentinel-2 (10m) 2015+ |
| Event Date | Date of storm, earthquake, weather event, etc. | 9 December 2016 |
| Pre-Event Window | The number of months to use for observing the greenest pixel-by-pixel conditions prior to the event | 12 |
| Post-Event Window | The number of months to use for observing the greenest pixel-by-pixel conditions following the event | 3 |
| Maximum Cloud Cover | The maximum percentage of a scene obscured by clouds and still used in the analysis. The cloud-cover percent is embedded in the metadata for each Landsat or Sentinel scene. | 30 |
| Slope Threshold | A minimum topographic slope value in degrees, less than which will be omitted from the data visualization. This is helpful to remove water bodies like lakes and adjacent oceans in coastal regions. | 0.01 |

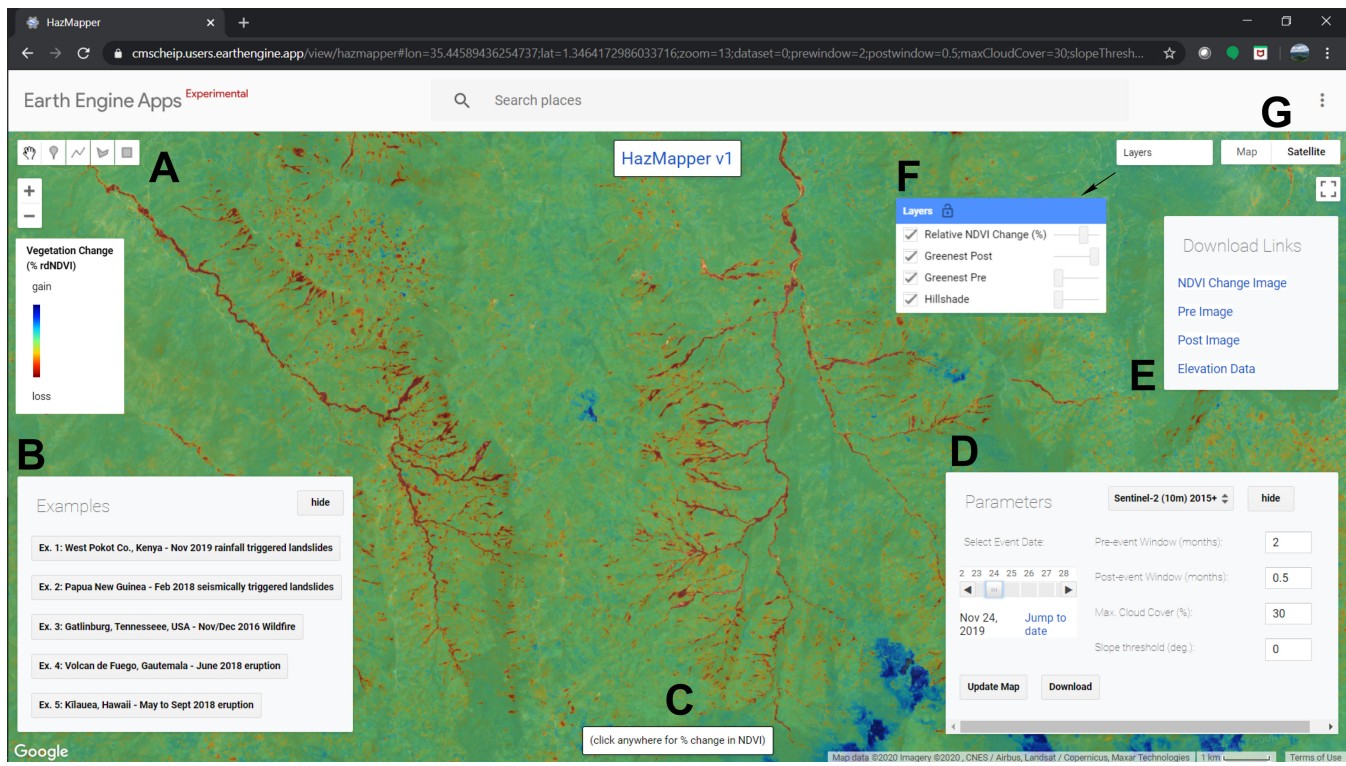

**Figure 1.** User interface of the HazMapper application in Google Earth Engine. Moving counterclockwise around the interface: A) Digitization tools allow the user to digitize features of interest as points, lines, or polygons. B) An example window allows users to explore five curated examples. Data on this map shows results from Example 1: November 2019 rainfall-triggered mass wasting, West Pokot County, Kenya. C) Users can display $rdNDVI$ results from a single point by clicking at a location on the map. The displayed result is in the format "(latitude, longitude), $rdNDVI$ result." D) The parameters window allows users to select the various input values, as further explained in Table 1 and section 4.2 of the Discussion. This panel also contains the Download button, which initiates the population of download links as seen in E. The 'Update Map' button will restart data processing following user modification of the event parameters. E) Download links for the resulting datasets allows users to save data directly to their local disk for further analysis and processing as desired. If any digitized geometries are present, a Digitized Geometries link will allow users to save a Google Earth compatible KML file. F) A Layers pane contains four consistent layers, including a hillshade DEM, the greenest composite pre- and post-event pixel images, and the $rdNDVI$ image. Users can choose to turn on/off each layer as well as adjust the layer transparency. G) The default basemaps available in Google Earth Engine include satellite imagery with or without labels and a standard borders map with or without terrain data. Notice that the URL reflects current HazMapper parameters. URLs are automatically updated during use of the app. This design feature facilitates sharing of finds in HazMapper amongst colleagues by simply copying and pasting the URL into an email, or instant message chat screen, for example.

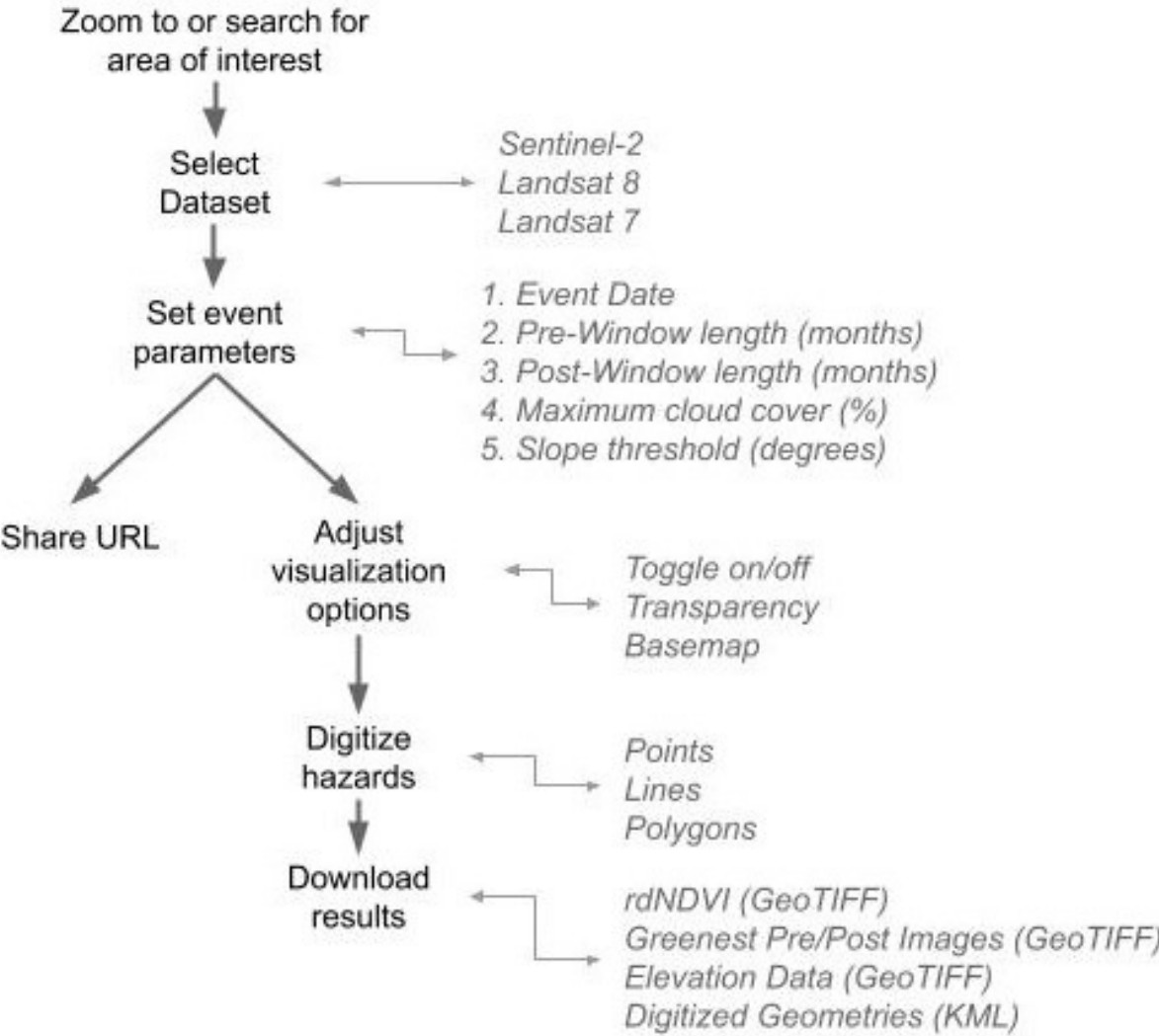

**Figure 2.** Suggested HazMapper workflows, including a branching point to share the URL or continue with analysis. Workflow culminates in downloading tagged image files (GeoTIFF) suitable for input into a GIS for advanced analysis or visualization functions. If the user digitizes key areas of interest, or hazards such as mass wasting processes, burn, or inundation extents, these can be exported as keyhole markup language (KML) files for sharing or viewing in Google Earth.

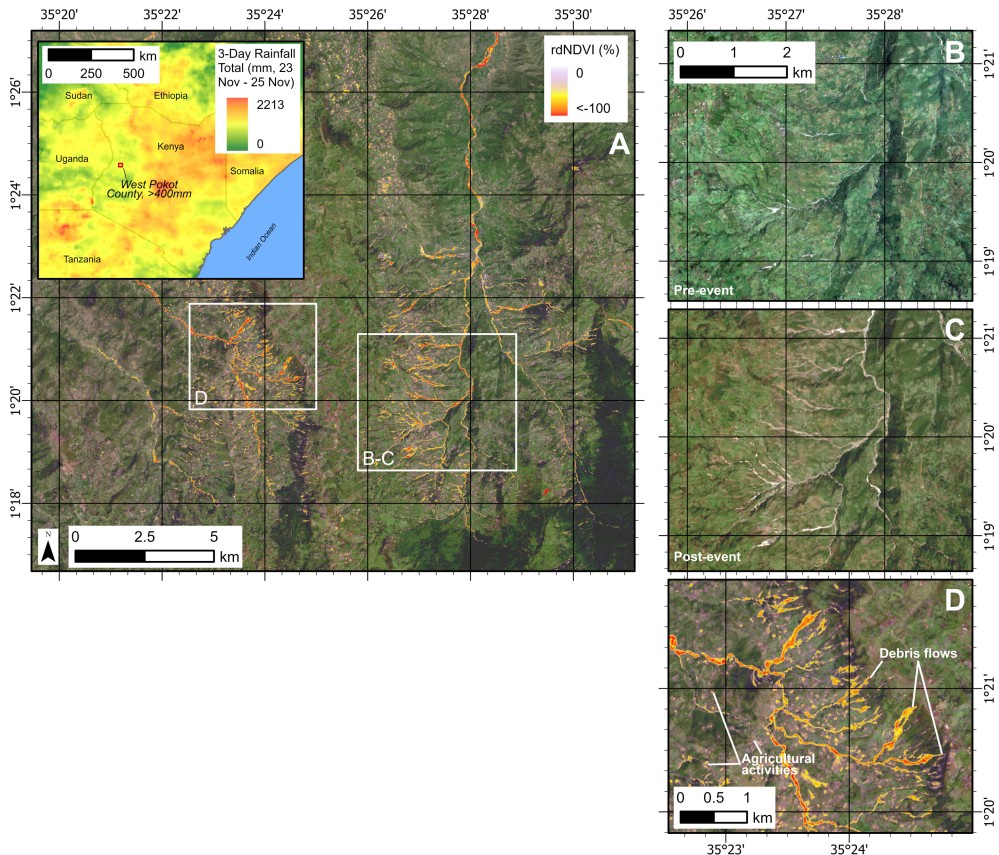

**Figure 3.** $rdNDVI$ change detection images and greenest pixel composites following the 23 November 2019 rainfall-induced mass wasting event in West Pokot County, Kenya. See inset map for location and 3-day rainfall totals from storm, courtesy of NASA Integrated Multi-satellitE Retrievals for GPM (IMERG) program. Parameters - Dataset: Sentinel-2, Event Date: 23 Nov 2019, Pre-Window: 2 months, Post-Window: 0.5 months, maximum cloud cover: 30%, and slope threshold: 0°. West Pokot County received >400 mm of rain in the 72-hour period 23-25 November, triggering landslides, debris flows, and floods during the heaviest rainfall on 23 November. A) $rdNDVI$ illustrates hillslope and low-order stream channel disturbance, leading to debris flows on the upper slopes and vegetation loss and sedimentation along river channels flowing toward the north and northwest. Landscape change is easier to interpret with $rdNDVI$ compared to observations based only on pre- and post-event color images (B and C). Base image is post-event. B) Pre-event greenest pixel composite image showing relatively green vegetative cover across landscape. C) 0.5-month post-event greenest pixel composite illustrates reduction of vegetation in landslide and debris flow tracks and along river trunk channel. This reduction in vegetation is noted by negative $rdNDVI$ values as seen in A. D) Close-up detail of negative $rdNDVI$ values associated with mass wasting. Rectilinear patches of negative $rdNDVI$ values in western area of panel D illustrate agricultural clearing or harvest activities across the event parameters. In mass wasting applications, further use of slope thresholding and interpretation based on landscape morphology will reduce these undesired rdNDVI artifacts. User interpretation of output $rdNDVI$ polygon areas is always warranted in order to minimize unwanted artifacts. B and C have the same map scale. A and D have the same $rdNDVI$ color-scale. $rdNDVI$ and greenest pixel composite data exported from HazMapper, available at https://go.ncsu.edu/hazmapper-kenya.

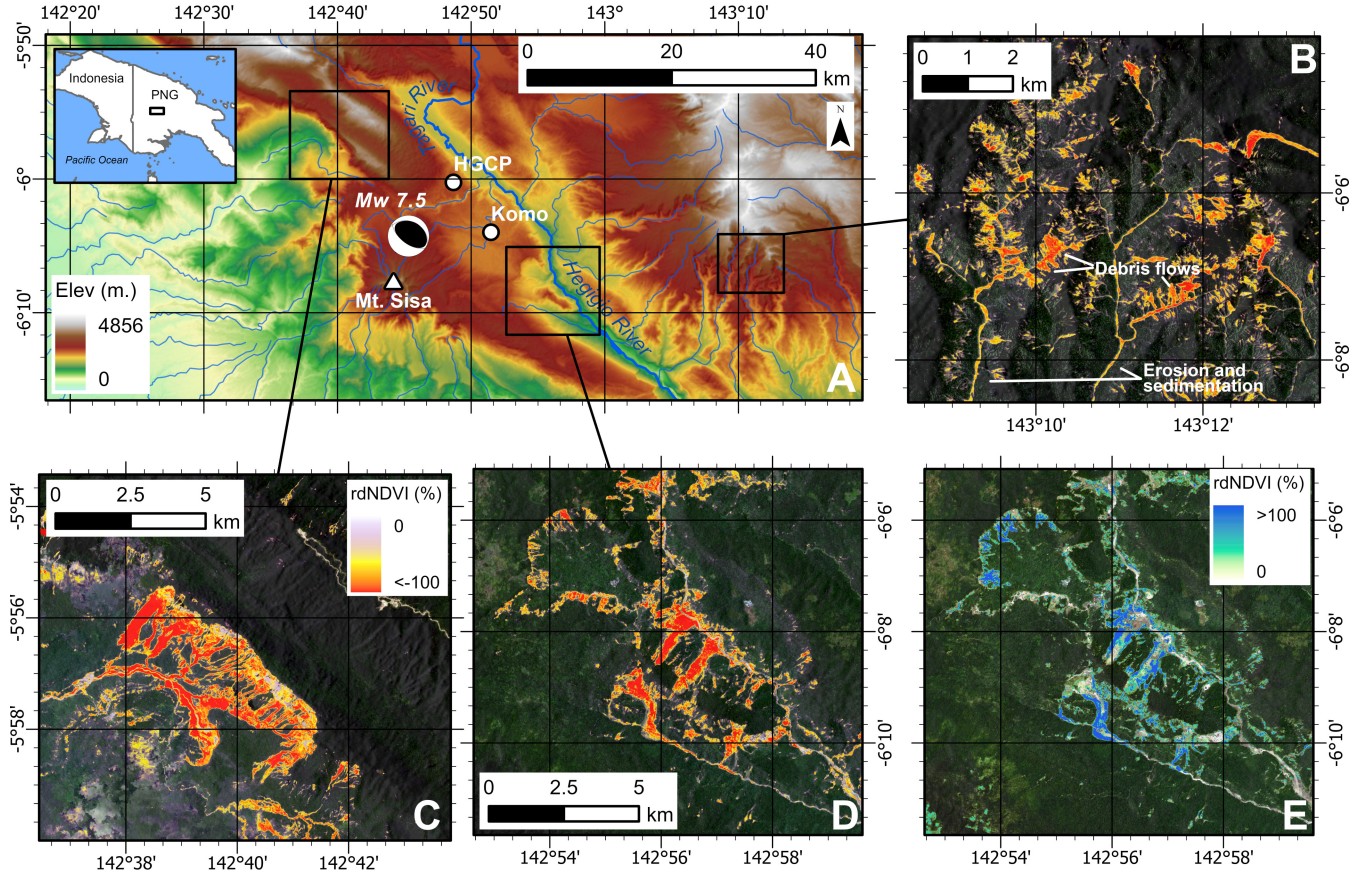

**Figure 4.** $rdNDVI$ change detection images and greenest pixel composites following a landsliding event triggered by the $M_w$ 7.5, 26 February, 2018 earthquake in the Southern Highlands of Papua New Guinea (PNG). Inset map shows location within the country. Parameters for A-D - Dataset: Sentinel-2, Event Date: 25 Feb 2018, Pre-Window: 12 months, Post-Window: 9 month, maximum cloud cover: 30%, and slope threshold: 0.05°. A) Elevation map for the Southern Highlands of PNG with epicenter (focal mechanism) plotted. HGCP = Hides Gas Conditioning Plant, an Exxon-Mobil liquefied natural gas facility. Komo is the nearest large town and Mt. Sisa is a stratovolcano to the south. B-D) Select areas with high mass wasting density. Base images are the pre-event greenest pixel composite and B-D have the same $rdNDVI$ color scale. E) Recovery change detection image illustrating increases in vegetation within areas of previous mass wasting. These increases in vegetation are expected to increase root mass and provide a stabilizing effect for exposed soils. Parameters - Dataset: Sentinel-2, Event Date: 26 Aug 2018, Pre-Window: 6 months, Post-Window: 6 month, maximum cloud cover: 30%, and slope threshold: 0.05°. $rdNDVI$ and greenest pixel composite data exported from HazMapper, available at https://go.ncsu.edu/hazmapper-png.

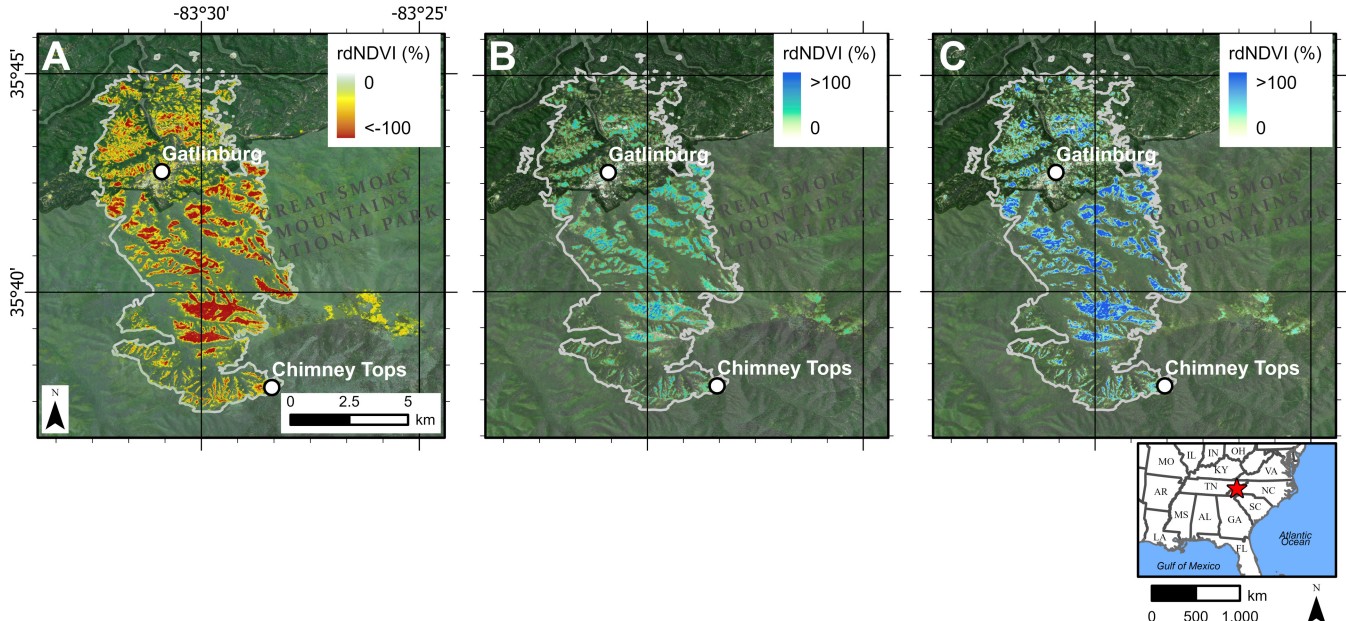

**Figure 5.** $rdNDVI$ change detection images of vegetation loss and recovery during and following the Chimney Tops 2 fire that impacted Gatlinburg, Tennessee, USA and surrounding communities in November - December, 2016 (see location map to right). A) $rdNDVI$ illustrates vegetative loss during the fire. Gray line is the 6,936 hectares (17,140 acre) published extent of the burn (USGS, 2020). Note the preferential vegetative loss (burn) along ridgetops. Base image is pre-event. Parameters - Event Date: 13 Dec 2016, Pre-Window: 12 months, Post-Window: 9 months. B) Change detection image of $rdNDVI$ illustrating vegetative recovery between the first (2017) and second (2018) growing seasons following the fire. Base image is post-event. Parameters - Event Date: 13 Dec 2017, Pre-Window: 12 months, Post-Window: 12 months. C) Vegetative recovery ($rdNDVI$) between the first (2017) and third (2019) growing seasons following the fire. Note the general pattern of continuation and expansion of re-greening of the landscape, indicated by generally higher $rdNDVI$ values (blues). Base image is post-event. Parameters - Event Date: 13 Dec 2017, Pre-Window: 12 months, Post-Window: 24 months. For all panels, Dataset: Sentinel-2, maximum cloud cover: 30%, and slope threshold: 0°. $rdNDVI$ and greenest pixel composite data exported from HazMapper, available from https://go.ncsu.edu/hazmapper-chimneytops.

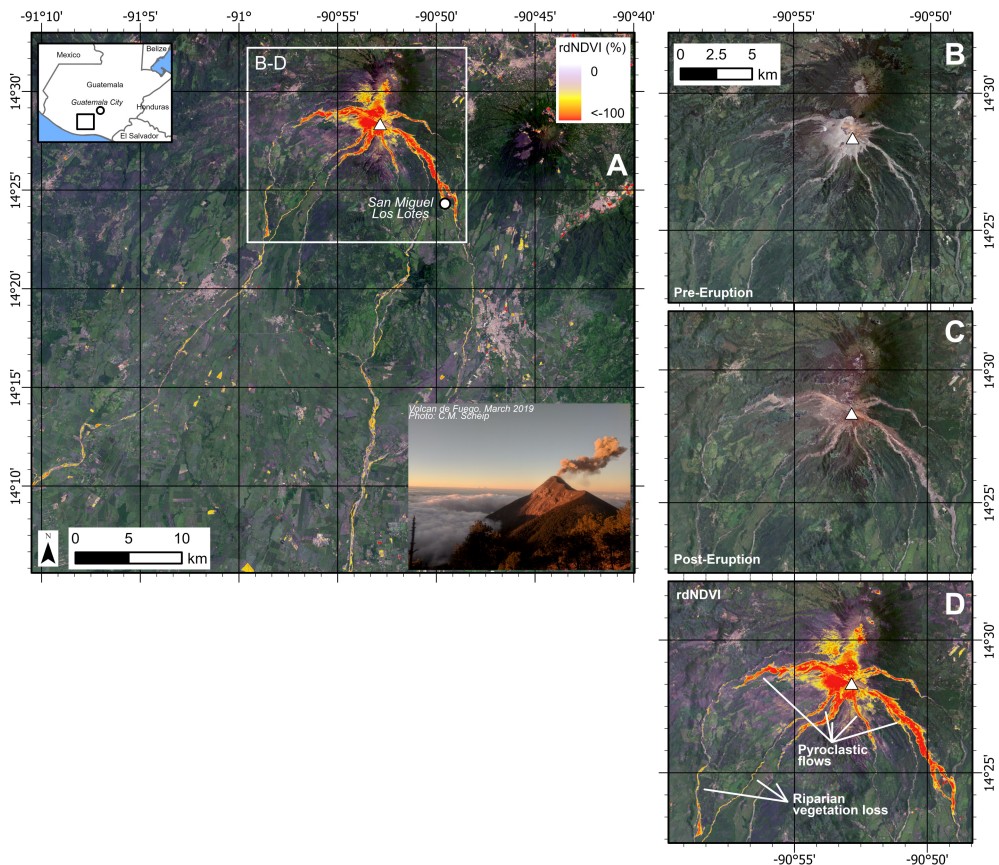

**Figure 6.** $rdNDVI$ change detection images and greenest pixel composites following the 3-5 June 2018 volcanic eruption of Volcan de Fuego 40 km southwest of Guatemala City, Guatemala. White triangle denotes summit. Parameters - Dataset: Landsat-8, Event Date: 3 June 2018, Pre-Window: 12 months, Post-Window: 3 months, maximum cloud cover: 30%, and slope threshold: 0.05°. A) $rdNDVI$ across the event illustrates loss of vegetation on upper flanks of the volcano summit and influence of pyroclastic flows on volcano flanks and downstream areas. San Miguel Los Lotes is a small community on the southeastern flanks of Volcan de Fuego that was heavily impacted during the eruption, including at least 25 fatalities. Note significant riparian vegetation loss in channels flowing south to southwest away from volcano. Base image is post-eruption. B-D) Close-up detail of Volcan de Fuego. B) Pre-eruption greenest pixel composite image of volcano. Note limited vegetation near summit. C) Detail image of volcano summit post-eruption greenest pixel composite. A typical color composite pre-post comparison can be performed to locate areas impacted by the eruption and resulting pyroclastic flows, however, $rdNDVI$ as shown in D provides a more rapid approach to identifying impacted areas and adds additional detail such as riparian vegetation loss that is more difficult to observe in a standard (R,G,B) color image comparison. D) $rdNDVI$ of the volcano following the eruption with notable observations annotated. B-D have the same scale. A and D use the same $rdNDVI$ color-scale. $rdNDVI$ and greenest pixel composite data exported from HazMapper, available at https://go.ncsu.edu/hazmapper-fuego.

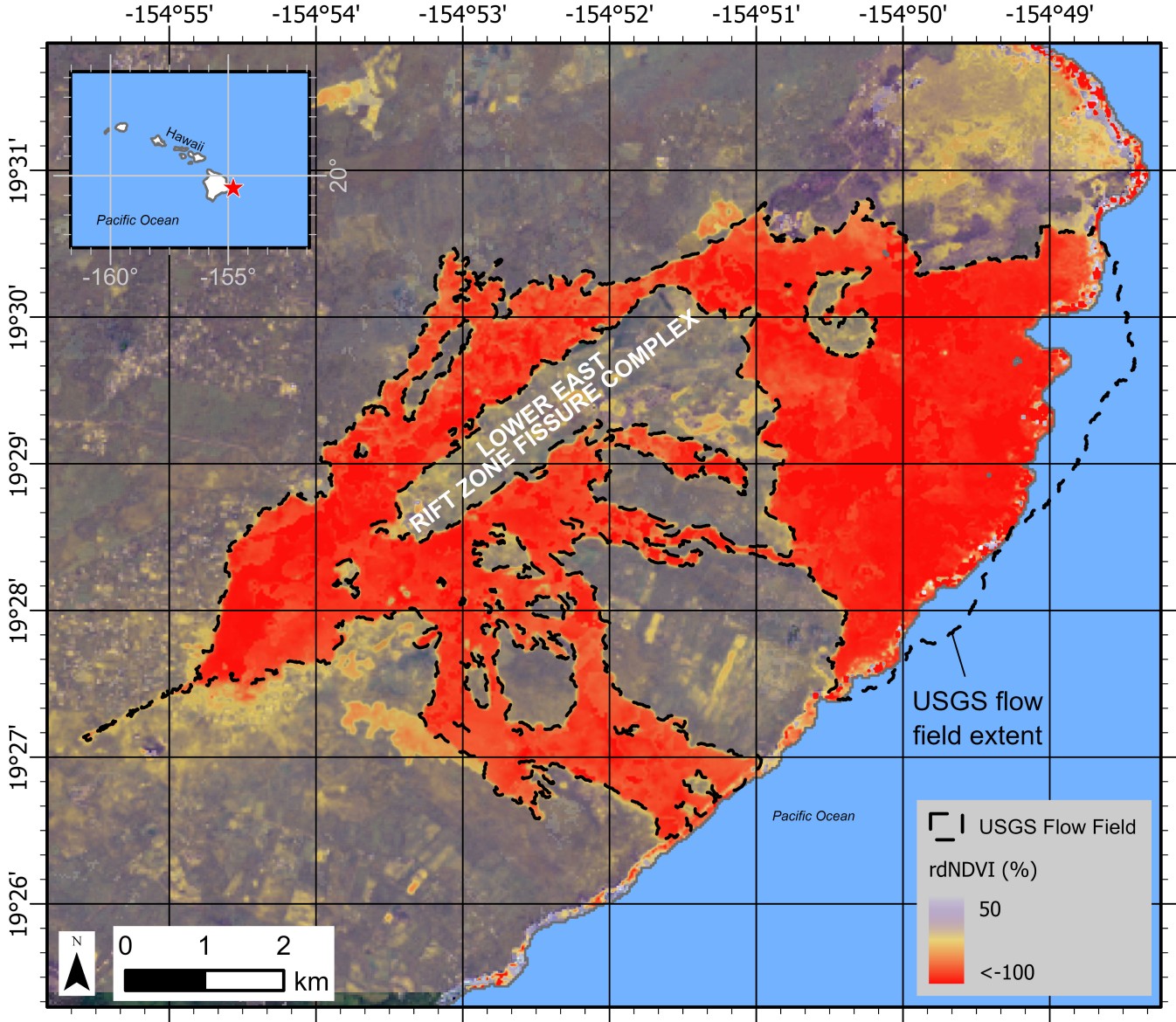

**Figure 7.** $rdNDVI$ change detection images and greenest pixel composites following 3 May - 4 September 2018 Lower East Rift Zone eruption of Kīlauea volcano, Hawaii, USA. Parameters - Dataset: Landsat-8, Event Date: 4 September 2018, Pre-Window: 12 months, Post-Window: 6 months, maximum cloud cover: 30%, and slope threshold: 0.05°. $rdNDVI$ across the event illustrates loss of vegetation. Base image is post-eruption. Black dashed line is the published extent of the lava flow field (Hawaiian Volcano Observatory staff, 2018) for comparison to the HazMapper result. Notice the additional land mass added to the island by the eruption that is encapsulated by the lava flow perimeter polygon. Because there was no vegetation in this area before and after the eruption, the $rdNDVI$ method does not account for the new landmass. In future analyses, however, we expect to be able to identify vegetation growth on the landmass. $rdNDVI$ and greenest pixel composite data exported from HazMapper, available at https://go.ncsu.edu/hazmapper-lerz.

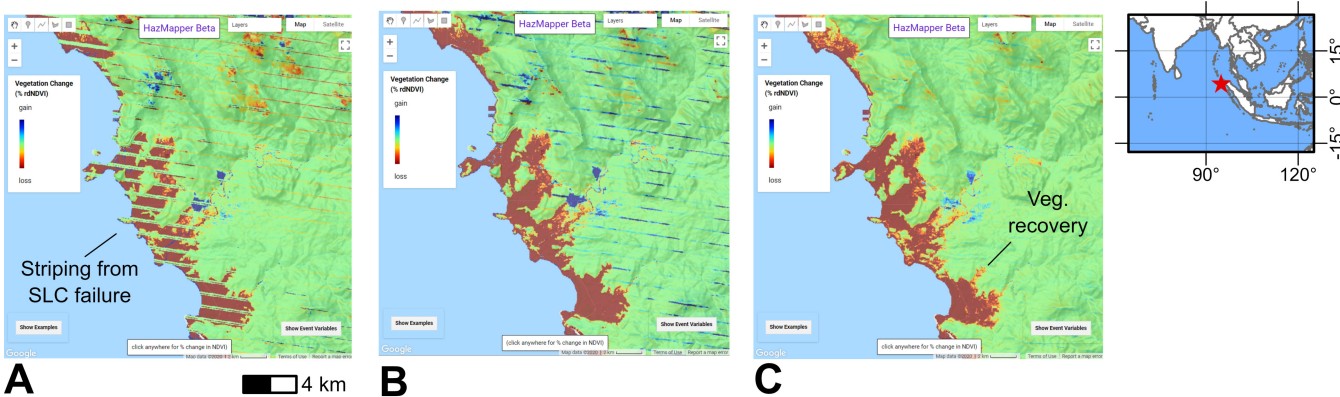

**Figure 8.** $rdNDVI$ change detection images illustrate coastal erosion, vegetation removal, and sedimentation following the 26 December 2004 tsunami in Indonesia resulting from the Sumatra–Andaman earthquake. Parameters - Dataset: Landsat-7, Event Date: 26 December 2004, maximum cloud cover: 100%, and slope threshold: 0.01°. Pre-Windows: 1, 2, and 12 months, Post-Window: 1, 2, and 12 months for panels A, B, and C, respectively. $rdNDVI$ across the event illustrates tsunami inundation zone and resulting loss in vegetation. Striping in the data from the scan-line corrector failure in Landsat-7 is evident in a short look window (e.g. 1 month, A), but these artifacts are reduced by increasing the pre- and post-event windows (B and C). By 12-month pre-post periods (C), the striping is significantly reduced in results, however, vegetative recovery is also present in this longer post-event cycle that captures the first growing season following the tsunami. This example is accessible at https://go.ncsu.edu/hazmapper-indonesia-tsunami.