# Peer review of "HazMapper: A global open-source natural hazard mapping application in Google Earth Engine"

_Natural Hazards and Earth System Sciences, 2020_

## Referee Comment (RC1) · Anonymous Referee #1 · 16 Jun 2020

Scheip and Wegmann present an online-GIS to display and analyse perturbations of the Earth's surface. This toolbox allows the user to select among three different satellite missions, to choose a period of interest, and calculate landscape changes using a vegetation index. The authors show five case studies to visualize the detectable impacts from volcanic, coseismic, and rainfall-related mass movements, and burnt areas from wildfires.

The authors argue that HazMapper is designed for people with little prior knowledge in a GIS environment, or who could have limited access to powerful computing facilities. This goal seems to be fulfilled given that the interface (Figure 1) is designed

in clear and visually appealing fashion. The downside of the presented web GIS is that the possibilities—for the time being—remain very limited beyond calculating a vegetation index. Clearly, practitioners may benefit from the resulting maps of vegetation change. Yet from a scientific perspective these maps need at least a minimum amount of quality check to judge how useful this maps are. Yet, unfortunately, any measure of accuracy or uncertainty (and discussion thereof) remains elusive in the current manuscript. Some questions (without logical sorting or relevance) that could be answered in more detail are: - What can we do in regions with frequent cloud cover?

- How can we detect mass movements that do not cause disruption of the vegetation cover, such as slowly moving landslides or mass movements in arid or un-vegetated (high mountain) regions?

- How does HazMapper perform in the era before 2012, when only patchy Landsat 7 images are available?

- How does the resolution of the sensor affect the minimum size of detected disturbances?

- Are rdNDVI values comparable across the three different sensors? What are the optimal thresholds to set during analysis?

- How can we make sure that the automatically detected changes come from the same trigger?

I highly appreciate the goal of the authors to help non-experts in doing rapid post-event analysis, but I found few information that guides these non-experts through their analysis. Limited knowledge about the regrowth rates, for example, could lead to large misestimates of detected changes, if the window is not set accordingly during the analysis. It was therefore surprising to see that the current manuscript offers no discussion section where such issues are considered in detail.

In the following, I list some minor line-by-line comments that mostly address possibilities to clarify or shorten the manuscript.

L14: Is it important to distinguish between 'developed' and 'undeveloped' countries here?

L17: What is 'significant' here?

L18: 'characterization': more specific?

L20: 'typically persist in vegetated landscapes': Could the authors briefly explain why that is the case?

L23-24: 'provides a single time-stamp of ground conditions': People familiar with dating would argue that one can read out way more from a landscape than a single time-stamp from one field visit. . .

L24: Why should people with no 'interest' perform field work?

L26-27: 'observe, monitor, and track': suggest using only one of these terms L29: How do the authors define 'increasingly complex'?

L35: 'obvious advantages': such that? Could there also be some 'not so obvious' disadvantages, for example in the field of data protection regulations?

L55: Use the more familiar 'GeoTIFF' instead 'geoTIF'?

L58: 'Can be' instead of 'is'?

L59: 'other opaque atmospheric components': such as? Maybe haze and dust?

L58-63: Not sure whether this motivation is useful at this stage, because all these arguments call for using non-optical data sets such as radar, given that they suffer less from atmospheric disturbances.

L64-69: Are the satellite images radiometrically corrected?

L98: Would be good if the curated examples are publicly accessible without having a

Google account.

L112: What if internet access is limited or unavailable in 'regions with less adequate resources'? Would this rule out the use of the HazMapper?

L112: Please avoid subjective terms such as 'incredibly'.

L113-115: These two sentences are slight repetitions from previous arguments, for example in L21-24. Consider shortening (or deleting).

L115-116 & L118-119: These arguments have also been brought up before. Could it make sense to redistribute the content of Chapter 3 into Chapters 1 and 2? Most of these argument could strengthen the overall motivation of this paper in the introduction. I do not see too much additional value in a chapter on its own that compares different GIS environments.

L128-135: These sentence largely contain arguments from previous sections, and could be more useful to expand the line of arguments (or number of references) there.

L134-L137: These two sentences should go into Section 2.

L138-139: Repetition, consider deleting.

L151-154: Again repetitions from previous sections. By the way, I'm not sure whether downloading 'one to a few pre- and post-event images' demands 'high-powered computers and large digital storage capacity'. The authors may acknowledge that downloading one Landsat scene before and one after a landslide, each ∼800 MB large, and loading them into memory can be done with the bulk of post-2010 computers, no?

L157, L159: What can Huffman's paper from 2014 tell us about a debris flow that happened in November 2019?

L165-173: What is the reason for this mini-review on the local geology / geomorphology? It feels like this paragraph dissects a bit the logical flow between the preceding and the following paragraph in this chapter.

L185: 'Fatalities from coseismic mass wasting events can increase significantly': from which baseline do fatalities increase and by which rate? And how do the authors define 'significantly' in this regard?

L197: 'when expanding the analysis window to the predicted 300 km maximum distance': What did the authors prevent from not considering the full radius of 300 km from the beginning?

L198-199: How did the authors make sure that these landslides were generated from the same earthquake?

L199-201: Structure of the sentence is not fully clear. Please elaborate, possibly splitting this sentence into two.

L201: 'future' is a bit odd for a tool that uses historic images. . .

L202: What is the content of this 'robust spatial and temporal catalog'? Figure 4 shows no more than a rdNDVI map, without explicitly digitizing individual landslides or debris flows, measuring their areas, estimating their volumes, their runout paths, spatial density, potential different time stamps of occurrence, and so on. Also, how do the authors define 'robust' here? So far, the authors show no accuracy assessment, in terms of how much of the total area (as one measure of accuracy) is correctly classified by the rdNDVI. What are the commission / omission errors, compared to manually mapped landslides? It is hard to believe that this approach picks mass wasting events error-free, especially in tropical regions with rapid regrowth rates along river channels.

L234: '17,000 acres': what would be the area that HazMapper predicts?

L237: If HazMapper only uses the rdNDVI, how can it distinguish between burnt areas and a landslide that happened in that area, or clear cutting?

L238-239: How do the authors measure 'the most severe burn', assuming there are different types of vegetation cover with a study region that might have completely different starting NDVI values?

L249: Is there a reference that shows that the annual number of fatalities from volcanic eruptions has increased from year to year in the past 500 years?

L254 / L264: 'downslope hazards': more specifically?

L265: What do the authors mean be 'decimated'?

L273: These 'analytical false negatives' urgently demand quantification!

L274: How can the authors judge from satellite images that these features are 'hyper-concentrated flows'? And how they define the 'transition to hyper-concentrated stream flows'?

L290: What do the authors recommend for cases where we have persistent cloud cover, possibly over months to years? There are many coastal and mountain regions, and many scientists or practitioners would wish to see a solution to this problem. This calls for a fuller discussion regarding the limitations of HazMapper.

L291: 'Future code modifications': such as? This could be a core problem of HazMapper: How can we map landscape change if there is no vegetation?

L297: And what does this 'comparison' show? Seismologists tend to use seismic data, and HazMapper uses optical satellite images, both parties probably have a hard time to make their datasets comparable to each other. Where is the overlap?

L303: No discussion chapter?

L317: How 'good' is this 'approximation' in real numbers?

Figures 3, 4, 6: What are the slope thresholds good for? Why are they (close to) zero?

Figure 4: Why do C and D have a different color scale? Could it be that D and E have wrong labels?

---

## Referee Comment (RC2) · Anonymous Referee #2 · 3 Aug 2020

The authors present a simple interface to Google Earth Engine that allows the user to map the impact of natural disasters using changes in vegetation seen in satellite images. Specifically, the relative difference in Normalized Difference Vegetation Index (rdNDVI) of Ambrose et al (2019) is calculated from Sentinel and Landsat images acquired before and after the natural disaster, and the resulting map of rdNDVI is displayed in such a way to highlight the vegetation changes. The interface then provides a tool to digitize these changes and export the resulting shape, along with the original images, to various formats. The impact of opaque atmospheric clouds is mitigated by the use of greenest pixel composites, which mosaic images over a given time period, selecting for each pixel the greenest values from the images in that period. The utility of

this approach is shown with 5 case studies, two of which are validated using published data. I have been able to follow the procedure outlined in the flow chart in figure 2 and confirm that the functionality works as presented.

The novelty in this paper is the combination of the pre existing methods of rdNDVI and greenest pixel compositing, along with the Google Earth Engine service, to map hazard related vegetation changes, as well as a simplified user interface to make this accessible to the public. The authors are to be commended for trying to span the gap between the world of cloud computing and big data, which allow very large satellite datasets to be processed rapidly in response to a crisis, and government agencies and the general public, who could gainfully interact with the data, but may not have the expertise to engage with it through the Google Earth Engine directly. The web interface is streamlined and easy to use, and the rdNDVI technique appears to work well in the case studies, although it would be nice to see more ground truthing, or at least some kind of verification for the examples where there is none. I believe HazMapper has the potential to be very useful in the future and the authors have succeeded in producing a useful tool.

My main criticisms are as follows:

(1) The change detection method presented is entirely based on vegetation changes. This is not mentioned in the title and abstract, so the scope of the paper is really quite a bit smaller than a cursory glance would imply.

(2) I'm not sure the website succeeds in making natural hazard impact assessment completely open to non-experts. It seems you still need to know something about green-pixel compositing and why there might still be 'holes' in the images that will then propagate into strange features in the rdNDVI images, as well as what rdNDVI is measuring and how vegetation regrowth might affect it. I don't think rdNDVI can simply be used as a naive index of change that non expert use of the system implies. I do think this can be remedied by adding a tutorial mode that walks you through the case

studies, pointing out what it all means and where the method fails. There are many JavaScript libraries that make this painless to implement, such as driver.js.

(3) Apart from Figure 1, the figures that present the results of using HazMapper in various settings seem to have been made in ArcMap using data exported from HazMapper – this gives a somewhat misleading presentation of what the website actually does. In particular, it does not seem to be possible to reproduce the style of presentation in figure 3, which looks like thresholded rdNDVI over greenest pixels in HazMapper. I think either this thresholding ability needs to be added to HazMapper (which shouldn't be hard) or the figures need to be changed to screenshots from HazMapper to reflect the actual user experience.

(4) Stylistically, the paper can be quite "wordy" with redundant words within sentences and the same concept being explained repeatedly in successive sentences and in successive sections of the paper. I feel the whole paper could do with being edited for brevity – I've highlighted some examples in the detailed comments below.

(5) The paper is quite "jumbled up". Advantages of Google Earth Engine are mixed in with the advantages of HazMapper, proven applications in the case studies are mixed in with proposals for future work, sentences extolling the virtues of HazMapper that read a bit like ad copy are present, some parts seem more like a research proposal. It's not always clear what's novel, what's been done before, and what is being proposed for the future. All of these components have their place in a paper (except the ad copy!), but they need to be separated out. I understand presenting a new tool like this departs from the traditional structure of a research paper, but I think the solution is to treat the case studies more like a results section, and have a traditional discussion section that assesses the results, and an extended conclusion section in which all the consideration for future work can be placed.

(6) It seems to me that only making HazMapper available to the reviewers during submission to an open review journal is a missed opportunity to get feedback from a wider

base of potential users. If the purpose of the paper is partly to advertise for potential users and contributors (as the current abstract implies), having it open at the review stage would seem like a good idea.

(7) The source code should have been made available to the reviewers. The text also does not explicitly mention what licence the code will be released under – this should be included.

(8) The web interface could also be improved with an rdNDVI color bar and distance / area measurement tools, if possible.

(9) The website can be quite slow, and it is not always immediately obvious that the little bars in the layers button show the loading time. Also. sometimes the little loading bars indicate loading is complete, and yet downloads are not available. If there is extra "loading time" required in the background after the images are displayed but before they are available for download, this should be indicated to the user in some way.

Detailed Comments:

Line 1: "rapid repeat-cycles" here the words "repeat" and "cycle" are effectively saying the same thing, and don't tell you what is being repeated. I would change to "rapid image acquisition cycles".

Line 7: "HazMapper is openly available to the public" repeats the claim of line 4. It only needs to be said once in the abstract.

Line 9: "It is the intent of the authors ..." this reads like an advertisement or community announcement more suitable for a conference or email list than the abstract of a research paper. I would remove it. And if the intent is to advertise for users during an open review, the software probably ought to be open at the time of review. Limiting access to the web page until after the review is complete seems like a missed opportunity. Plus the source code itself really should be available to the reviewers.

Line 14: "developed . . . undeveloped" this distinction doesn't seem to be pertinent to

the following discussion, I would remove.

Line 20: Again, not sure how this sentence is pertinent to the following discussion – this seems to be a point about a spatial scale below which natural disasters don't leave a lasting mark in vegetated landscapes resulting in incomplete historical records, but this point doesn't seem to be pursued? The following sentence seems to be saying that organizations are looking for evidence of these events, despite them leaving "no readily observable field evidence" in most instances, which doesn't make a lot of sense. I would rephrase this.

Line 23: "field work is inefficient" compared to what? Maybe change to "inefficient compared to remote sensing methods" or something like this.

Line 23: "provides a single time-stamp" - perhaps change to "snapshot in time", or some other wording – I think one can say data is timestamped, but I'm not sure it makes sense to say that a timestamp is provided, unless you are referring to a digital (or physical) text string that gives the time and date that is then attached to a bit of data?

Line 25: "The advent of rapid-repeat cycle satellite datasets …. has revolutionized" → maybe change to just "revolutionized", or cut entirely. "has revolutionized" implies a recent or ongoing revolution, but as you point out in the next sentence the "revolution" is now almost 50 years old, and monitoring environmental change with high res satellites is now a pretty orthodox thing to do. However, we are in the middle of a micro satellite resolution, with one provider (Planet) aiming to cover the entire globe everyday, currently at a much higher frequency than Landsat/sentinel with a much higher spatial resolution and NDVI capability. This is not mentioned in this paper, and it is obviously commercial in nature, but it should be noted that this is a rival platform with similar ambitions and capability, that is set to grow in the future.

Line 28: "subsequent satellite networks (e.g. MODIS, Sentinel ..." - MODIS isn't a satellite, maybe change to "subsequent sensors aboard different satellite constellations" or

suchlike

line 38: "scientific curious public" → "scientifically curious"?

Line 41: "HazMapper is useful for monitoring landscape change that results in disruption of surface vegetation" this seems an important point, as it is the entire basis of how HazMapper currently works, and should probably be in the abstract, and maybe the title. The authors point out their intent to add other approaches, both by their own efforts and through growing an online base of contributing developers, but that is for the future and not a research result being presented here.

Line 42: "While the underlying mathematics are not entirely novel ..." my understanding is that the mathematics presented here is not novel, and the novelty comes from the combination of technologies and how they are made available? I would delete this line, unless I've missed some novel mathematics somewhere, in which case it should be made explicit what exactly that is.

Line 43: "HazMapper democratizes ..." I'm wondering if democratizing is the appropriate word here – doesn't that imply some kind of collective decision making? I imagine HazMapper could be used in such a way by a group of people, but it doesn't seem to provide anything explicit to facilitate "democratic" decision making. It seems to me to be more about accessibility than anything else.

Line 50: "Because HazMapper is intended to be an emergency management tool . . . it is designed around user input variables" aren't all user interfaces formulated around user-input variables? I would skip this sentence.

Line 51: "Variables include ..." → something like: "The user is able to control the following variables ... and cloud cover" - make an exhaustive list.

Line 53: "Basemap options include . . . " → "The basemap options are"

Line 54: Missing "The" at start of sentence.

Line 58: Sentence starting "Optical aerial..." duplicates point in next sentence and can be removed.

Line 65: I assume the greenest pixel technique is something that comes built in to Google Earth engine? If so, is there a reference for it?

Line 74: Might be an idea to explicitly state what "pixel fractionation" is

Line 87: "Available basemaps on the platform include" repeating information from line 53? If so, delete.

Line 89: Is "Heads-up" digitization different from regular digitization?

Line 96: "For this article and shared finds ..." Not obvious what "shared finds" are – please rephrase.

Lines 106 -126: Section 3 is titled "Earth Engine vs. Traditional GIS Environments". However in this section it's not entirely clear if the virtues being extolled are those of HazMapper or Earth Engine. E.g. , lines 122-126 – both of these sentences apply to Google Earth Engine alone if one is proficient with Javascript and the GEE API? Would be good to rewrite this and make more explicit what the advantages of GEE are, and what HazMapper builds on top of that.

Lines 107-112: This is quite a long winded way of saying that the main aim of HazMapper is to make satellite image analysis available to less wealthy areas. Consider condensing into a sentence or two, rather than a paragraph. Plus it occurs to me that Google Earth Engine has already solved the problem of making modern scientific analysis available - it seems to me anyone with the computational resources to use HazMapper can use Google Earth Engine as well. Isn't HazMapper really about expertise, you're trying to make it available to those who aren't going to learn JavaScript and the GEE API for whatever reason?

Line 117: "timestamp" → "take a snapshot of"? The method uses greenest pixel composites, which seem fundamentally diachronous in nature, whereas "timestamp" to me

implies an instant in time.

Line 121: "HazMapper's source code" - > "The source code for HazMapper..." or "The HazMapper source code...". Plus the source code should really be available in an online repository and linked to from the paper. Also, what license will it be made available available under?

Line 129: "... difficult to overstate ..." I would rephrase to be less effusive. Also, reference seems to be for radar, which has different concerns to optical sensors, which are the subject matter here?

Line 138: "HazMapper is intended to facilitate future research..." This seems to be repeating the sentiment from the end of the introduction – I would expand that section rather than repeating here. Unless maybe you mean to say this future research will be added to the case studies, in which case I would say that explicitly here.

Line 143: "$" → "USD" or whatever the appropriate currency code is (assuming US Dollar is the currency).

Line 146: "Significant research" significant in what sense? Needs explanation.

Line 152: "...analysis by trained professionals with access to high-powered computers and large storage capacity..." this point has been made before, I would remove this sentence.

Line 152: "Whether to provide ... " This reads a bit more like advertising copy than scientific journal text, would consider rephrasing. Plus I think this has been said before in the paper, just in a general natural hazards context, rather than in a specific context (mass wasting). Maybe best to remove.

Line 178: "Acquisition schedules ..." This seems like a general statement about potential use of HazMapper, rather than something that was a part of this particular case study and should be in the discussion or conclusions section as part of a "potential use" section. It seems to me that the case studies function as a kind of results section in

this paper, and should be limited to those results, rather than speculation on un-proven, future usage.

Line 185: "Fatalities ... from the earthquake itself (Budimir et al., 2014" → "Fatalities from co seismic mass wasting can be up to an order of magnitude greater than fatalities resulting from the earthquake itself (Budimir et al., 2014"

Line 188: "... are not well understood for this event that occurred ..." →"are not well understood on account of this event occurring in a rural and remote ...."

Line 201: "HazMapper provides ....evaluation of empirical relationships between parameters such as moment magnitude ..." Seems to me this is about future work and so is out of the scope of the case studies presented here which function as a kind of results section of the paper. Again move to a "potential future work" section in the discussions or conclusions.

Paragraph starting 205: again mixing up future work and case studies. This part feels a bit like a research proposal.

Line 224: Again, referencing future potential – these aren't really results of the case studies and should be moved to the discussion.

Line 243: Future potential again.

Line 290: "advancement" → "advance", "met with obscurity" → "hindered"

Line 306: "democratize" see previous comments.

Line 312: "...suggesting the extents..." this seems to be the core of what the approach presented here does, and it needs to be highlighted more.

Line 314. "... rapid response ..." it seems to me the appropriate timescale for something to be considered a "rapid response" for a natural hazard, at least as far as non-academics are going to be concerned, is the timescale on which people's lives are lost as a result of the event and its aftermath, and two weeks is actually quite a long time

in this context?

Line 316: verification of maps produced by HazMapper could form part of a discussion section?

Line 355: "See letter to the editors" - is this appropriate to have under "code and data availability", and will the letter be available to readers when the manuscript is finally published? Should any important information in the letter be moved into this section?

Equations 1 and 2 – do equations in this journal have captions like figures? If not, I would replace the captions with text in the main article, something like "where x is the variable for blah, y is the rate of blablah, and z is the magnitude of blahblahblah".

Figure 1. "Heads up digitize" - again, is this different from regular digitization?

Figure 3. "All maps have same orientation" - unnecessary, delete – add N arrows to figs b,c,d if worried about ambiguity.

I was unable to reproduce the maps in this figure in HazMapper– it seems as if perhaps the rdNDVI is being thresholded somehow and overlain on the greenest pixel images in ArcMap? If this is the case, it is a bit misleading to be presenting this as an example of the successful use of HazMapper. Presumably it would be possible to add such a facility to HazMapper with relatively little effort, otherwise the figures should really be replaced with screenshots from HazMapper, (or at the very least it needs to be made more explicit that this is a more "advanced" analysis performed for validation purposes only, and that the facility will not be available to non-expert users within the HazMapper interface itself).

---

## Author Comment (AC1) · 4 Sep 2020

**Response to anonymous reviewer / RC1**

We sincerely thank the anonymous reviewer for thoughtful feedback on this manuscript, including both broad topical suggestions and grammatical or word choice suggestions. We especially appreciate these reviews considering this paper presents the first iteration of a new research tool.

A recurring misconception with the review is that HazMapper is a semi-automated method and that we did not provide quantifiable data needed to assess the rigor of a semi-automated approach in identifying natural hazard features. This initial release of the HazMapper application does not semi-automate the identification of natural hazard features. Identification of natural hazard features and assessment of the signal-to-noise ratio in the rdNDVI output is incumbent upon the user. At present, we believe many in the natural hazards research, prevention, and outreach communities will find HazMapper to be a useful utility that will assist in their exploration and characterization of many different types of natural hazard features. Our hope is that future iterations of the tool will focus on individual hazard types (e.g., mass wasting; wildfires; tornadoes) where further analysis and semi-automation will be employed.

Color legend
RC1 - black
Author response - blue

**Topical comments**
Scheip and Wegmann present an online-GIS to display and analyse perturbations of the Earth's surface. This toolbox allows the user to select among three different satellite missions, to choose a period of interest, and calculate landscape changes using a vegetation index. The authors show five case studies to visualize the detectable impacts from volcanic, coseismic, and rainfall-related mass movements, and burnt areas from wildfires. The authors argue that HazMapper is designed for people with little prior knowledge in a GIS environment, or who could have limited access to powerful computing facilities. This goal seems to be fulfilled given that the interface (Figure 1) is designed in clear and visually appealing fashion. The downside of the presented web GIS is that the possibilities for the time being remain very limited beyond calculating a vegetation index. Clearly, practitioners may benefit from the resulting maps of vegetation change. Yet from a scientific perspective these maps need at least a minimum amount of quality check to judge how useful this maps are. Yet, unfortunately, any measure of accuracy or uncertainty (and discussion thereof) remains elusive in the current manuscript. Some questions (without logical sorting or relevance) that could be answered in more detail are:

Response:
Thank you for the compliments on the web interface. For this initial release of HazMapper, a multi-spectral imagery index is calculated and no semi-automation is employed. That is, this iteration of HazMapper is not a semi-automated tool for identifying individual landslides or burn extents, for example. Instead, a vegetation index is displayed which the user can interpret as to the nature of the landscape change.

In its current state, we believe this tool is useful for a wide variety of researchers without semi-automation. As such, uncertainty is not quantified. Qualitatively, published burn extents and lava inundation extents from the Chimney Top 2 fire and Kilauea example are compared. We agree that uncertainty assessments should follow semi-automation where predictions of disaster occurrence are made but posit that this is premature for the current iteration of the platform.

What can we do in regions with frequent cloud cover?

Response:
From the reviewed manuscript, lines 64-65:
*To circumvent potentially opaque atmospheric conditions, HazMapper capitalizes on a technique within Google Earth Engine to generate and perform calculations on a greenest-pixel composite (Figure 2).*

and lines 66-67:
*...records the pixel with the highest normalized difference vegetation index (NDVI) result, or the "greenest" pixel (Eq. 1).*

We will expand on this text to make this more clear. We composite many images together and retain only the pixels with the highest NDVI value from the entire stack. This composite indicates the peak phenological cycle of pre- and post-event conditions. The rdNDVI metric is computed from this peak phenological cycle calculated from the user input duration (time - months) for the pre-and-post event windows.

"How can we detect mass movements that do not cause disruption of the vegetation cover, such as slowly moving landslides or mass movements in arid or un-vegetated (high mountain) regions?"

Response:
We will add text to the manuscript to reinforce that this is a vegetation-based metric and will not be useful in completely un-vegetated regions (e.g. desert, polar). Arid or humid, the key is the presence or absence of vegetation. For example, this summer, we have had success

tracking wildfires across the arid western United States, for example, the Bush Fire north of Phoenix, AZ [https://twitter.com/HazMapper/status/1277988549741207552; https://en.wikipedia.org/wiki/Bush_Fire_(Arizona)]. While this area is quite arid, vegetation is still burning, and, thus, we can register that loss with HazMapper. If a disaster does not impact vegetation, in most cases, HazMapper will not be a suitable tool for detecting it.

"How does HazMapper perform in the era before 2012, when only patchy Landsat 7 images are available?"

Response:
This is an excellent question! We will add text and a figure (draft figure included below) to the manuscript to discuss this. When using Landsat 7 data, short pre- and post-event windows will tend to return outputs with stripes. However, by increasing the window lengths, these artifacts are reduced via our greenest pixel compositing methods described in the manuscript.

[Figure]

**Figure 8.** $rdNDVI$ change detection images and greenest pixel composites following 26 December 2004 tsunami in Indonesia resulting from the Sumatra–Andaman earthquake. Parameters - Dataset: Landsat-7, Event Date: 26 December 2004, maximum cloud cover: 100%, and slope threshold: 0.01°. Pre-Windows: 1, 2, and 12 months, Post-Window: 1, 2, and 12 months for panels A, B, and C, respectively. $rdNDVI$ across the event illustrates tsunami inundation zone and resulting loss in vegetation. Striping in the data from the scan-line corrector failure in Landsat-7 is evident in a short look window (e.g. 1 month, A), but these artifacts are reduced by increasing the pre- and post-event windows (B and C). By 12-month pre-post periods (C), the striping is significantly reduced in results, however, vegetative recovery is also present in this longer post-event cycle that captures the first growing season following the tsunami. Screenshots of HazMapper application, example accessible at https://go.ncsu.edu/hazmapper-indonesia-tsunami.

How does the resolution of the sensor affect the minimum size of detected disturbances?

Response:
Sensor resolution is one variable affecting the minimum size of detected disturbances, finer resolution can detect finer disturbances and should be considered in the analysis. For example, in North Carolina, where the average landslide width is ~10m, Sentinel-2 would be more suitable for analysis compared to Landsat.

Are rdNDVI values comparable across the three different sensors? What are the optimal thresholds to set during analysis?

Response:
No, they are not, nor are they expected to be. Differences in pixel size (e.g. 30m vs. 10m) will result in different NDVI values, and therefore, rdNDVI values. Further, no rdNDVI thresholding is available in HazMapper. This is something that the user could choose to do after downloading data for further analysis.

How can we make sure that the automatically detected changes come from the same trigger?

Response:
The pre- and post-windows restrict data used for analysis. Therefore, changes can be confidently constrained to the time frame of the selected pre-post analysis window. This type of analysis provides more confidence in timing compared to traditional field visits that may occur weeks to months after an event and may not be accompanied with strong confidence in pre-event conditions. Further, this provides at least the same level of confidence as traditional single image comparison studies. In those studies, for example, a landslide present in the post-event image but not the pre-event image is typically assumed to have occurred during the event under consideration.

I highly appreciate the goal of the authors to help non-experts in doing rapid post-event analysis, but I found few information that guides these non-experts through their analysis. Limited knowledge about the regrowth rates, for example, could lead to large misestimates of detected changes, if the window is not set accordingly during the analysis. It was therefore surprising to see that the current manuscript offers no discussion section where such issues are considered in detail.

Response:
Thank you for this excellent perspective. Our initial thought was the event by event discussion captured much of what a traditional Discussion section would, however, the suggestion to add a Discussion section is well received and the revised manuscript will include a Discussion section.

No training materials have been developed yet, however, we have begun work on https://hazmapper.org/learn where we anticipate providing YouTube videos, cookbook examples, and other training materials for exactly this purpose. We do not feel this type of media content is suitable to submit for publication as a research article.

**Grammatical and technical suggestions such as word changes or further explanations required**

"L14: Is it important to distinguish between 'developed' and 'undeveloped' countries here?"
Response:
We will remove differentiation between developed and undeveloped countries.

L17: What is 'significant' here?
Response:
We will remove word 'significant'

L18: 'characterization': more specific?
Response:
Characterization of natural hazards such as the number or spatial distribution of landslides, progression, and final burn extent of a wildfire, for two examples. We will add clarifying text to the manuscript.

L20: 'typically persist in vegetated landscapes': Could the authors briefly explain why that is the case?
Response:
We will add "due to the constant regrowth cycle of vegetation in temperate environments". For example, a landslide scar in the humid highlands of Papua New Guinea will become covered with vegetation in the subsequent growing seasons (see Figure 4D).

L23-24: 'provides a single time-stamp of ground conditions': People familiar with dating would argue that one can read out way more from a landscape than a single time-stamp from one field visit.
Response:
We will remove the sentence.

L24: Why should people with no 'interest' perform field work?
Response:
Our intent was to suggest limited agency budgets often force project prioritization. With competing interests vying for agency funds, only those of high enough priority, or interest, can be investigated with resource-and-time-intensive methods like fieldwork. Additionally, in the time of a global pandemic, many universities and public science and land management agencies are minimizing, discouraging, or not allowing travel by their employees to minimize exposure risk to the COVID-19 virus. We will remove the word "interest".

L26-27: 'observe, monitor, and track': suggest using only one of these terms
Response:
We posit these have different meanings in the remote sensing literature and would like to leave as is. For example, we may observe vegetation change resulting from a wildfire long after the fire burns, but during the burn, we may be monitoring the fire for expansion, or we may be tracking the progression of the fire once it begins to expand.

L29: How do the authors define 'increasingly complex'?
Response:
We will replace "complex" with "advanced" to indicate that new satellites and payloads are more advanced than older platforms, for example, increased resolution, precision, and capability.

L35: 'obvious advantages': such that? Could there also be some 'not so obvious' disadvantages, for example in the field of data protection regulations?"
Response:
We will remove the word "obvious". We will need further clarification on what data protection regulations are the concern of RC1. HazMapper is a view-only platform at present, in that users view data but are not currently loading any data into Google Earth Engine.

L55: Use the more familiar 'GeoTIFF' instead 'geoTIF'?
Response:
We will replace geoTIF with GeoTIFF in manuscript.

L58: 'Can be' instead of 'is'?
Response:
We will replace "is" with "can be" in manuscript.

L59: 'other opaque atmospheric components': such as? Maybe haze and dust?
Response:
We will include "atmospheric aerosols" as this term is more inclusive than haze or dust, which refers to dry atmospheric particles. Aerosols also include liquid droplets like water vapor.

L58-63: Not sure whether this motivation is useful at this stage, because all these arguments call for using non-optical data sets such as radar, given that they suffer less from atmospheric disturbances"
Response:
The intent of this comment is to set up the next paragraph where we explain how the compositing method partially overcomes the optical data limitations.

L64-69: Are the satellite images radiometrically corrected?
Response:
Datasets currently used in HazMapper are corrected to Top of Atmosphere (TOA).

L98: Would be good if the curated examples are publicly accessible without having a Google account
Response:
We agree fully with this comment. The entire HazMapper platform, including using the web interface, the curated examples, and the source code, will be released with the initial publication to users with or without a Google account.

L112: What if internet access is limited or unavailable in 'regions with less adequate resources'? Would this rule out the use of the HazMapper?
Response:
Yes, this is a web-based tool and cannot be used without the internet.

L112: Please avoid subjective terms such as 'incredibly'.
Response:
We will remove subjective terms such as "incredibly." Thank you for the suggestion.

L113-115: These two sentences are slight repetitions from previous arguments, for example in L21-24. Consider shortening (or deleting)."
Response:
We will delete this repetition.

L115-116 & L118-119: These arguments have also been brought up before. Could it make sense to redistribute the content of Chapter 3 into Chapters 1 and 2? Most of these argument could strengthen the overall motivation of this paper in the introduction. I do not see too much additional value in a chapter on its own that compares different GIS environments.
Response:
This is a great suggestion. We will remove the existing Chapter 3 *Earth Engine vs. Traditional GIS Environments*. That content is better suited for Chapters 1 and 2, as suggested by RC1.

L128-135: These sentence largely contain arguments from previous sections, and could be more useful to expand the line of arguments (or number of references) there.
Response:

We will move this text and use it to expand the number of references, as suggested by RC1.

L134-L137: These two sentences should go into Section 2.
Response:
We will move this text to Section 2, as suggested by RC1.

L138-139: Repetition, consider deleting.
Response:
Repetition will be deleted.

L151-154: Again repetitions from previous sections. By the way, I'm not sure whether downloading 'one to a few pre- and post-event images' demands 'high-powered computers and large digital storage capacity'. The authors may acknowledge that downloading one Landsat scene before and one after a landslide, each ~800 MB large, and loading them into memory can be done with the bulk of post-2010 computers, no?
Response:
Repetition will be deleted. Yes, modern computers can download "one to a few images" but it becomes increasingly difficult to download and process many dozens of images or to expand spatial extent after processing static images. Furthermore, HazMapper works on mobile devices like Chromebooks, tablets, and smartphones, which in most cases do not have the required software nor processing speed to perform a similar analysis. We have added text to this effect in the (new) Discussion section.

L157, L159: What can Huffman's paper from 2014 tell us about a debris flow that happened in November 2019?
Response:
We are happy to modify this database reference and agree it can be confusing. We are following the publisher's' citation recommendation (available at https://gpm.nasa.gov/data/policy), which suggests a 2014 citation year, consistent with the database release, and including a note of the access date, which in our case, reads 2020-01-24.

L165-173: What is the reason for this mini-review on the local geology / geomorphology? It feels like this paragraph dissects a bit the logical flow between the preceding and the following paragraph in this chapter.
Response:
We will modify this text to make this read a bit more logically. Thank you for the suggestion.

L185: 'Fatalities from coseismic mass wasting events can increase significantly': from

which baseline do fatalities increase and by which rate? And how do the authors define 'significantly' in this regard?

Response:

We removed the word "significantly."

L197: 'when expanding the analysis window to the predicted 300 km maximum distance': What did the authors prevent from not considering the full radius of 300 km from the beginning?

Response:

It can be difficult to view small landslides when viewing 300km of data, so we start the analysis zoomed in, and then expand out to identify total impacts.

L198-199: How did the authors make sure that these landslides were generated from the same earthquake?

Response:

Lines 198-201 of the reviewed manuscript reply to this comment:

*Furthermore, we noted possible coseismic slides and flows as far as several hundred km west of the epicenter in the Maoke Mountains of Indonesia. Mass wasting is common in the region and these events could have unique triggers, however, restricting pre- and post-event time windows to as little as 2 months bracketing the Mw 7.5 mainshock demonstrates consistent timing with the 25 February 2018 earthquake.*

We mention "possible" coseismic slides and flows, and agree that they "could have unique triggers." However, we can be confident that they occurred within the period or two months before the mainshock to two months after the mainshock because of restrictions in the pre- and post-event window lengths. Please let us know if additional clarification is required.

L199-201: Structure of the sentence is not fully clear. Please elaborate, possibly splitting this sentence into two.

Response:

We will divide this into two sentences, thank you for the suggestion.

L201: 'future' is a bit odd for a tool that uses historic images. . .

Response:

We agree with this comment and will remove"future" and re-word this sentence.

L202: What is the content of this 'robust spatial and temporal catalog'? Figure 4 shows no more than a rdNDVI map, without explicitly digitizing individual landslides or debris flows, measuring their areas, estimating their volumes, their runout paths, spatial density, potential different time stamps of occurrence, and so on. Also, how do the authors define 'robust' here? So far, the authors show no accuracy assessment, in terms of how much of the total

area (as one measure of accuracy) is correctly classified by the rdNDVI. What are the commission / omission errors, compared to manually mapped landslides? It is hard to believe that this approach picks mass wasting events error-free, especially in tropical regions with rapid regrowth rates along river channels.
Response:
We will remove this sentence. HazMapper does not "pick mass wasting events" as it is not currently a semi-automated method. Because it's not picking anything, evaluating commission/omission errors is premature. This type of analysis will be necessary once we implement the semi-automation of the platform, which will be hazard-specific and not a broad platform like this iteration of HazMapper.

L234: '17,000 acres': what would be the area that HazMapper predicts?
Response:
HazMapper does not "predict" a burn area because it is not a semi-automated method. The gray line in Figure 5 is the USGS-published burn extent, which measures 17,140 acres.

L237: If HazMapper only uses the rdNDVI, how can it distinguish between burnt areas and a landslide that happened in that area, or clear cutting?
Response:
We agree that non-unique solutions exist. For example, agricultural artifacts are highlighted on Figure 3D (Kenya debris flow example). As HazMapper is not a semi-automated method, at present the suggested interpretation (e.g. burned areas vs. landslide vs. clear-cutting) would be incumbent on the user.

L238-239: How do the authors measure 'the most severe burn', assuming there are different types of vegetation cover with a study region that might have completely different starting NDVI values?
Response:
From equation 2 in the reviewed manuscript, the starting NDVI value is used in the normalization parameter to account for exactly this phenomenon. The more severe burn is the highest rdNDVI value, consistent with Norman and Christie, 2020 (formerly Ambrose et al., 2019). This will be further clarified in the discussion section.

L249: Is there a reference that shows that the annual number of fatalities from volcanic eruptions has increased from year to year in the past 500 years?
Response:
We will add the Auker et al. (2013) reference to the text. This reference demonstrates what the reviewer requests regarding an increase in volcanic-hazard fatalities in the past 500 years. We specifically refer the reviewer to figure 4 of the Auker et al. (2013) paper, copied below.

[Figure]

**Figure 4 Time series of number of fatal incidents.** Blue line shows number of fatal incidents over time; red line shows 25-year moving average of number of fatal incidents over time. Counts are calculated in five-year cohorts.

L254 / L264: 'downslope hazards': more specifically?
Response:
The next sentence in the reviewed manuscript (line 255) defines downslope hazards associated with volcanoes:

*Downslope hazards may include lava flows, ballistic projectiles, pyroclastic flows, and lahars (Blong, 1984).*

Please let us know if additional clarification is needed.

L265: What do the authors mean be 'decimated'?
Response:
We will replace the word "decimated" with "destroyed."

L273: These 'analytical false negatives' urgently demand quantification!
Response:
The reviewed manuscript mentions noise as a concern (e.g. cloud cover, agricultural fields). This is another example of unwanted artifacts or noise. We recognize we should not have used the terms false negative or false positive, which would only apply in a predictive or semi-automated tool. We will revise the text to make this more clear.

L274: How can the authors judge from satellite images that these features are 'hyperconcentrated flows'? And how they define the 'transition to hyper-concentrated stream flows'?

Response:

The post-event differentiation between hyper-concentrated stream flows and debris flows is difficult to make even in the field, let alone remotely, we recognize. We do not suggest the exact location of this transition because it is unknown, as RC1 indicates. Based on the decreasing stream gradient away from the volcano flanks, at some point, it is a reasonable assumption the transition will occur, and we try to simply state that. Certainly the debris flow conditions do not persist for 60km across gentle slopes near the Pacific Ocean, for example. We will add clarifying text to explain this.

L290: What do the authors recommend for cases where we have persistent cloud cover, possibly over months to years? There are many coastal and mountain regions, and many scientists or practitioners would wish to see a solution to this problem. This calls for a fuller discussion regarding the limitations of HazMapper.

Response:

This problem and the solution is described earlier in the reviewed manuscript:

*To circumvent potentially opaque atmospheric conditions, HazMapper capitalizes on a technique within Google Earth Engine to generate and perform calculations on a greenest-pixel composite (Figure 2). The greenest pixel composite is a single composite image generated from all images within the user-defined pre- and post-event window that records the pixel with the highest normalized difference vegetation index (NDVI) result, or the "greenest" pixel (Eq. 1).*

We will expand on this in the revised manuscript to add additional clarification and will discuss this issue again in the (new) Discussion section.

As a practical example of this, the reviewer is invited to assess our example from Papua New Guinea (Figure 4), where the problem of persistent cloud cover is a hindrance to remote sensing for natural hazard applications. HazMapper leverages the greenest-pixel composite method discussed in the manuscript to reduce or remove cloud cover from scenes.  In some locations, compositing over several months may be required in order to derive a cloud-free greenest pixel composite image for either the pre or post-event window. In the case of the 2018 earthquake in PNG, we composited 12 months of pre-event and 9 months of post-event imagery in order to derive figures 4B, C, and E.

L291: 'Future code modifications': such as? This could be a core problem of HazMapper: How can we map landscape change if there is no vegetation?

Response:

We have initiated collaborative work with others on non-vegetation based metrics, but agree it is premature to include this sentence. We will remove it. We further agree that in lieu of vegetation, a vegetation based metric is not suitable. We make no claim that HazMapper will work in every environment and will ensure the revised manuscript indicates it is a vegetation-based metric that only applies to areas with vegetation.

L297: And what does this 'comparison' show? Seismologists tend to use seismic data, and HazMapper uses optical satellite images, both parties probably have a hard time to make their datasets comparable to each other. Where is the overlap?
Response:
The comparison is qualitative and is shown in Figure 7, where the USGS-published lava inundation extents are plotted against the rdNDVI results. Because HazMapper is not a semi-automated routine, no quantitative comparison is appropriate.

L303: No discussion chapter?
Response:
This is an excellent suggestion and the revised manuscript will include a Discussion section.

L317: How 'good' is this 'approximation' in real numbers?
Response:
We will change the language to represent the qualitative approximation.

Figures 3, 4, 6: What are the slope thresholds good for? Why are they (close to) zero?
Response:
We refer the reviewer to Table 1, which defines the slope threshold and suggests its use.

| Slope Threshold | A minimum topographic slope value in degrees, less than which will be omitted from the data visualization. This is helpful to remove water bodies like lakes and adjacent oceans in coastal regions. |
| --- | --- |

In the examples included in the manuscript, a very low slope threshold is useful to omit water bodies (e.g. oceans). Text will be added to the Discussion section to clarify this.

Figure 4: Why do C and D have a different color scale? Could it be that D and E have wrong labels?"
Response:
The labels are correct. C and D are different color scales because C is a figure depicting vegetation loss (negative rdNDVI values ) and D is a vegetative gain figure (positive rdNDVI values). Neither D nor E is labeled incorrectly. We will review the caption to ensure it clearly explains the differences.

---

## Author Comment (AC2) · 4 Sep 2020

**Response to anonymous reviewer / RC2**

We sincerely thank the anonymous reviewer for thoughtful feedback on this manuscript, including both broad topical suggestions and grammatical or word choice suggestions. We especially appreciate these reviews considering this paper presents the first iteration of a new research tool.

A recurring misconception with the review is that HazMapper is a semi-automated method and that we did not provide quantifiable data needed to assess the rigor of a semi-automated approach in identifying natural hazard features. This initial release of the HazMapper application does not semi-automate the identification of natural hazard features. Identification of natural hazard features and assessment of the signal-to-noise ratio in the rdNDVI output is incumbent upon the user. At present, we believe many in the natural hazards research, prevention, and outreach communities will find HazMapper to be a useful utility that will assist in their exploration and characterization of many different types of natural hazard features. Our hope is that future iterations of the tool will focus on individual hazard types (e.g., mass wasting; wildfires; tornadoes) where further analysis and semi-automation will be employed.

Color legend
RC1 - black
Author response - blue

**Topical comments**
The authors present a simple interface to Google Earth Engine that allows the user to map the impact of natural disasters using changes in vegetation seen in satellite images. Specifically, the relative difference in Normalized Difference Vegetation Index (rdNDVI) of Ambrose et al (2019) is calculated from Sentinel and Landsat images acquired before and after the natural disaster, and the resulting map of rdNDVI is displayed in such a way to highlight the vegetation changes. The interface then provides a tool to digitize these changes and export the resulting shape, along with the original images, to various formats. The impact of opaque atmospheric clouds is mitigated by the use of greenest pixel composites, which mosaic images over a given time period, selecting for each pixel the greenest values from the images in that period. The utility of this approach is shown with 5 case studies, two of which are validated using published data. I have been able to follow the procedure outlined in the flow chart in figure 2 and confirm that the functionality works as presented. The novelty in this paper is the combination of the pre existing methods of rdNDVI and greenest pixel compositing, along with the Google Earth Engine service, to map hazard related vegetation changes, as well as a simplified user interface to make this accessible to the public. The authors are to be commended for trying to span the gap

between the world of cloud computing and big data, which allow very large satellite datasets to be processed rapidly in response to a crisis, and government agencies and the general public, who could gainfully interact with the data, but may not have the expertise to engage with it through the Google Earth Engine directly. The web interface is streamlined and easy to use, and the rdNDVI technique appears to work well in the case studies, although it would be nice to see more ground truthing, or at least some kind of verification for the examples where there is none. I believe HazMapper has the potential to be very useful in the future and the authors have succeeded in producing a useful tool.

Response:
Thank you for the compliments on the web interface and thank you for testing HazMapper.

The change detection method presented is entirely based on vegetation changes. This is not mentioned in the title and abstract, so the scope of the paper is really quite a bit smaller than a cursory glance would imply.

Response:
We will add language in the abstract and manuscript to clarify this point.

I'm not sure the website succeeds in making natural hazard impact assessment completely open to non-experts. It seems you still need to know something about green-pixel compositing and why there might still be 'holes' in the images that will then propagate into strange features in the rdNDVI images, as well as what rdNDVI is measuring and how vegetation regrowth might affect it. I don't think rdNDVI can simply be used as a naive index of change that non expert use of the system implies. I do think this can be remedied by adding a tutorial mode that walks you through the case studies, pointing out what it all means and where the method fails. There are many JavaScript libraries that make this painless to implement, such as driver.js

Response:
Driver.js is an outstanding suggestion! We will explore that for http://hazmapper.org/learn. We have also initiated work with a large European university to develop a landslide mapping module with them this autumn using HazMapper.

Apart from Figure 1, the figures that present the results of using HazMapper in various settings seem to have been made in ArcMap using data exported from HazMapper – this gives a somewhat misleading presentation of what the website actually does. In particular, it does not seem to be possible to reproduce the style of presentation in figure 3, which looks like thresholded rdNDVI over greenest pixels in HazMapper. I think either this thresholding ability needs to be added to HazMapper (which shouldn't be

hard) or the figures need to be changed to screenshots from HazMapper to reflect the actual user experience.

Response:
We have tested some user-driven color ramps in HazMapper and will implement in the future, but for the first iteration, believe a generic -50 to +50% change in rdNDVI red to blue color ramp best suits most users. This color ramp saturates at -50 and +50%. Values greater than this will have the same color at a 50% decrease or gain in rdNDVI. The download functionality of HazMapper is intended for users to download data for use in other geographic information systems software platforms (e.g., ArcGIS, QGIS, GRASS GIS, etc.,). We have posted our ArcGIS Pro compatible symbology files used to render the data presented in the figures on our website: https://hazmapper.org/resources-faq/

Additionally, a new figure 8 (included at the end of this document) to address a comment from RC1 about the utility of Landsat 7 data and HazMapper because of it's well known scan-line error issues will include the default map and color ramp layout that a user would experience at the HazMapper user interface.

Stylistically, the paper can be quite "wordy" with redundant words within sentences and the same concept being explained repeatedly in successive sentences and in successive sections of the paper. I feel the whole paper could do with being edited for brevity – I've highlighted some examples in the detailed comments below

Response:
Thank you for this comment. We agree with the assessment and will remove repetitive portions of the text.

The paper is quite "jumbled up". Advantages of Google Earth Engine are mixed in with the advantages of HazMapper, proven applications in the case studies are mixed in with proposals for future work, sentences extolling the virtues of HazMapper that read a bit like ad copy are present, some parts seem more like a research proposal. It's not always clear what's novel, what's been done before, and what is being proposed for the future. All of these components have their place in a paper (except the ad copy!), but they need to be separated out. I understand presenting a new tool like this departs from the traditional structure of a research paper, but I think the solution is to treat the case studies more like a results section, and have a traditional discussion section that assesses the results, and an extended conclusion section in which all the consideration for future work can be placed.

Response:

We will move the content of Chapter 3 Google Earth Engine vs. Traditional GIS environments into Chapters 1 and 2. This is consistent with a similar comment from RC1 and is well received. We will reduce "ad copy". We will include a Discussion section.

It seems to me that only making HazMapper available to the reviewers during submission to an open review journal is a missed opportunity to get feedback from a wider base of potential users. If the purpose of the paper is partly to advertise for potential users and contributors (as the current abstract implies), having it open at the review stage would seem like a good idea.

Response:
We have included several peers in the HazMapper Reviewers Google Group and they have been user-testing the application on various projects - academic research, applied emergency management work, and academic instruction. Coupled with this peer-review, our intent was to gather feedback from established professionals and researchers prior to a public release.

The source code should have been made available to the reviewers. The text also does not explicitly mention what licence the code will be released under – this should be included.

Response:
If the reviewers would like a copy of the source code, please let us know and we are happy to send it. License information for the source code will be included in the revised version of the manuscript.

The web interface could also be improved with an rdNDVI color bar and distance / area measurement tools, if possible

Response:
We have implemented a rdNDVI color bar - you will see this if you log in to HazMapper again (screenshot below). Thank you for the suggestion. Distance/area measurement tools are a bit harder to implement and we have added to our app-improvements suggestion list. For now, users can use the default scale bar (lower right) to estimate distances. Of course, by downloading data or digitizing and downloading features, further measurements can be made.

[Figure]

*Example of new rdNDVI color bar, Mass Wasting induced by Cyclone Idai, March 2019, Zimbabwe/Mozambique border region*

The website can be quite slow, and it is not always immediately obvious that the little bars in the layers button show the loading time. Also. sometimes the little loading bars indicate loading is complete, and yet downloads are not available. If there is extra "loading time" required in the background after the images are displayed but before they are available for download, this should be indicated to the user in some way.

Response:
You have described the exact behavior of trying to analyze too large of an area. Even Google Earth Engine has limitations, unfortunately. A good example is the 2019 Australia brush fires, which are difficult to analyze because doing so requires processing 10m (Sentinel-2) or 30m (Landsat) data at the continental scale.

If you have no download link appearing, you are trying to download too large of an area. Google Earth Engine currently limits downloads/exports to ≤32MB. We will describe this limitation in the text and add a warning message to the HazMapper interface to indicate such. We will continue to monitor the development of Google Earth Engine for increases to this file-size limit. Additionally, we are exploring ways to export data to a Google Drive if desired, which increases these download limits but also decreases accessibility to the application.

**Grammatical and technical suggestions such as word changes or further explanations required**

Line 1: "rapid repeat-cycles" here the words "repeat" and "cycle" are effectively saying the same thing, and don't tell you what is being repeated. I would change to "rapid image acquisition cycles".

Response:
We will remove this language from line 1 as suggested, but have left it elsewhere in the manuscript. The term "repeat cycle" is consistent with the remote sensing literature and the agencies responsible for launching and maintaining the main satellites used in HazMapper ESA: https://earth.esa.int/web/guest/missions/esa-operational-eo-missions/sentinel-2
USGS:
https://www.usgs.gov/land-resources/nli/landsat/landsat-8?qt-science_support_page_related_con=0

Line 7: "HazMapper is openly available to the public" repeats the claim of line 4. It only needs to be said once in the abstract

Response:
We will remove the 2nd occurrence, thank you for the suggestion.

Line 9: "It is the intent of the authors ..." this reads like an advertisement or community announcement more suitable for a conference or email list than the abstract of a research paper. I would remove it. And if the intent is to advertise for users during an open review, the software probably ought to be open at the time of review. Limiting access to the web page until after the review is complete seems like a missed opportunity. Plus the source code itself really should be available to the reviewers.

Response:
We apologize for the misinterpretation of this statement and will remove it from the manuscript.

We have included several peers in the HazMapper Reviewers Google Group and they have been user-testing the application on various projects - academic research, applied emergency management work, and academic instruction. Coupled with this peer-review, our intent was to gather feedback from established professionals and researchers prior to a public release.

Regarding the source code availability, please see our response above to a similar comment.

Line 14: "developed : : : undeveloped" this distinction doesn't seem to be pertinent to the following discussion, I would remove.

Response:
We will remove this distinction from the manuscript.

Line 20: Again, not sure how this sentence is pertinent to the following discussion – this seems to be a point about a spatial scale below which natural disasters don't leave a lasting mark in vegetated landscapes resulting in incomplete historical records, but this point doesn't seem to be pursued? The following sentence seems to be saying that organizations are looking for evidence of these events, despite them leaving "no readily observable field evidence" in most instances, which doesn't make a lot of sense. I would rephrase this

Response:
Field evidence of vegetation disturbance from natural disaster events is ephemeral - in humid environments, vegetation grows back rapidly. Using a time-series of satellite data helps to look back in time even after vegetation has regrown, which is something we cannot do with fieldwork alone. We will modify the manuscript to clarify this point. Thank you for the suggestion.

Line 23: "field work is inefficient" compared to what? Maybe change to "inefficient compared to remote sensing methods" or something like this.

Response:
That is correct, compared to the remote sensing methods. We will modify the manuscript to clarify this.

Line 23: "provides a single time-stamp" - perhaps change to "snapshot in time", or some other wording – I think one can say data is timestamped, but I'm not sure it makes sense to say that a timestamp is provided, unless you are referring to a digital (or physical) text string that gives the time and date that is then attached to a bit of data?

Response:
We will modify the manuscript to remove "time-stamp".

Line 25: "The advent of rapid-repeat cycle satellite datasets : : :. has revolutionized" ! maybe change to just "revolutionized", or cut entirely. "has revolutionized" implies a recent or ongoing revolution, but as you point out in the next sentence the "revolution" is now almost 50 years old, and monitoring environmental change with high res satellites is now a pretty orthodox thing to do. However, we are in the middle of a micro satellite resolution, with one provider (Planet) aiming to cover the entire globe everyday, currently at a much higher

frequency than Landsat/sentinel with a much higher spatial resolution and NDVI capability. This is not mentioned in this paper, and it is obviously commercial in nature, but it should be noted that this is a rival platform with similar ambitions and capability, that is set to grow in the future.

Response:
We will remove the word "has" from line 25.

We are also very excited about the current micro-sat revolution. However, we respectfully disagree that a private corporation like Planet Labs is a "rival platform" or that the corporation has "similar ambitions" to a government-funded open access operation such as Landsat or Sentinel.

One current solution could include building an advanced HazMapper platform suited for use within the Google Earth Engine Python or JavaScript API, which allows users to bring in external datasets for analysis (such as those they purchased from Planet or another commercial provider). This is significantly outside the scope of the current HazMapper application but could be considered in the future.

Line 28: "subsequent satellite networks (e.g. MODIS, Sentinel ..." - MODIS isn't a satellite, maybe change to "subsequent sensors aboard different satellite constellations" or suchlike

Response:
We will modify the manuscript to read "networks *and payloads*" to capture the MODIS payload.

line 38: "scientific curious public" ! "scientifically curious"?

Response:
We will accept this suggestion and modify the manuscript.

Line 41: "HazMapper is useful for monitoring landscape change that results in disruption of surface vegetation" this seems an important point, as it is the entire basis of how HazMapper currently works, and should probably be in the abstract, and maybe the title. The authors point out their intent to add other approaches, both by their own efforts and through growing an online base of contributing developers, but that is for the future and not a research result being presented here

Response:
We will clarify this point in the abstract.

Line 42: "While the underlying mathematics are not entirely novel ..." my understanding is that the mathematics presented here is not novel, and the novelty comes from the combination of technologies and how they are made available? I would delete this line, unless I've missed some novel mathematics somewhere, in which case it should be made explicit what exactly that is.

Response:
We will remove the word "entirely" from the manuscript. We want to ensure the reader understands our contribution is not the mathematics or multispectral satellite index implementation, but the combination of technologies as the reviewer points out.

Line 43: "HazMapper democratizes ..." I'm wondering if democratizing is the appropriate word here – doesn't that imply some kind of collective decision making? I imagine HazMapper could be used in such a way by a group of people, but it doesn't seem to provide anything explicit to facilitate "democratic" decision making. It seems to me to be more about accessibility than anything else.

Response:
"Data democratization" is a common term in the data science and technology sectors. This is the idea of making data available and accessible to everyone. As such, we respectfully disagree with this comment and prefer the current language.

Line 50: "Because HazMapper is intended to be an emergency management tool, it is designed around user input variables" aren't all user interfaces formulated around user-input variables? I would skip this sentence

Response:
The interactive nature of HazMapper is unique in remote sensing data science specifically because variables are not hard-coded and analysis is not performed for fixed spatial extents, but instead, these are user-input variables. We prefer to leave this language as is.

Line 51: "Variables include ..." ! something like: "The user is able to control the following variables ... and cloud cover" - make an exhaustive list.

Response:
We will accept this suggestion and modify the manuscript.

Line 53: "Basemap options include : : : " ! "The basemap options are"

Response:
We will accept this suggestion and modify the manuscript.

Line 54: Missing "The" at start of sentence.

Response:
We will accept this suggestion and modify the manuscript.

Line 58: Sentence starting "Optical aerial..." duplicates point in next sentence and can be removed.

Response:
We will accept this suggestion and modify the manuscript.

Line 65: I assume the greenest pixel technique is something that comes built in to Google Earth engine? If so, is there a reference for it?

Response:
Yes, it is built into GEE. There is no peer-reviewed reference for this particular function, but additional documentation is available here:
https://developers.google.com/earth-engine/apidocs/ee-imagecollection-qualitymosaic

Line 74: Might be an idea to explicitly state what "pixel fractionation" is

Response:
We will clarify this topic in the manuscript.

Line 87: "Available basemaps on the platform include" repeating information from line 53? If so, delete.

Response:
We will accept this suggestion and modify the manuscript.

Line 89: Is "Heads-up" digitization different from regular digitization?

Response:
Yes, it is different from manual digitizing (http://wiki.gis.com/wiki/index.php/Digitizing), however, we concur that in the modern computational environment, heads up digitizing is becoming the norm. To this end, we will accept this suggestion and modify the manuscript.

Line 96: "For this article and shared finds ..." Not obvious what "shared finds" are – please rephrase.

Response:
Shared finds was intended to describe natural hazards that other users located via HazMapper. We will revise the manuscript to clarify this.

Lines 106 -126: Section 3 is titled "Earth Engine vs. Traditional GIS Environments".However in this section it's not entirely clear if the virtues being extolled are those of HazMapper or Earth Engine. E.g. , lines 122-126 – both of these sentences apply to Google Earth Engine alone if one is proficient with Javascript and the GEE API? Would be good to rewrite this and make more explicit what the advantages of GEE are, and what HazMapper builds on top of that.

Response:
We will redistribute and clarify items from Section 3 to Section 1 and 2, removing Section 3.

Lines 107-112: This is quite a long winded way of saying that the main aim of HazMapper is to make satellite image analysis available to less wealthy areas. Consider condensing into a sentence or two, rather than a paragraph. Plus it occurs to me that Google Earth Engine has already solved the problem of making modern scientific analysis available - it seems to me anyone with the computational resources to use HazMapper can use Google Earth Engine as well. Isn't HazMapper really about expertise, you're trying to make it available to those who aren't going to learn JavaScript and the GEE API for whatever reason?

Response:
Google Earth Engine requires coding knowledge and expertise (JavaScript in the Code Editor, JavaScript API, or Python API) and an approved application to Google for becoming a "Google Earth Engine Developer". We are offering our expertise so that others may use an rdNDVI utility without going through the steps of learning to code or applying to the Google Earth Engine Developer program. We will review the manuscript for wordiness.

Line 117: "timestamp" ! "take a snapshot of"? The method uses greenest pixel composites, which seem fundamentally diachronous in nature, whereas "timestamp" to me implies an instant in time

Response:
We will remove the sentence.

Line 121: "HazMapper's source code" - > "The source code for HazMapper..." or "The HazMapper source code...". Plus the source code should really be available in an online repository and linked to from the paper.

Response:

We will accept this suggestion and modify the manuscript. The URL for the source code is listed in line 333 and will be posted in an online repository.

Also, what license will it be made available available under?

Response:
License information will be provided in the revised version of the manuscript.

Line 129: "difficult to overstate ..." I would rephrase to be less effusive. Also, reference seems to be for radar, which has different concerns to optical sensors, which are the subject matter here?"

Response:
We will remove this paragraph from the manuscript.

Line 138: "HazMapper is intended to facilitate future research..." This seems to be repeating the sentiment from the end of the introduction – I would expand that section rather than repeating here. Unless maybe you mean to say this future research will be added to the case studies, in which case I would say that explicitly here.

Response:
We will remove this paragraph from the manuscript.

Line 143: "$", "USD" or whatever the appropriate currency code is (assuming US Dollar is the currency).

Response:
We will accept this suggestion and modify the manuscript.

Line 146: "Significant research" significant in what sense? Needs explanation

Response:
We will remove "significant" from the manuscript.

Line 152: "...analysis by trained professionals with access to high-powered computers and large storage capacity..." this point has been made before, I would remove this sentence

Response:
We will accept this suggestion and modify the manuscript.

Line 152: "Whether to provide ... " This reads a bit more like advertising copy than scientific journal text, would consider rephrasing. Plus I think this has been said before in the paper, just in a general natural hazards context, rather than in a specific context (mass wasting). Maybe best to remove.

Response:
We will accept this suggestion and modify the manuscript.

Line 178: "Acquisition schedules ..." This seems like a general statement about potential use of HazMapper, rather than something that was a part of this particular case study and should be in the discussion or conclusions section as part of a "potential use" section. It seems to me that the case studies function as a kind of results section in this paper, and should be limited to those results, rather than speculation on un-proven, future usage.

Response:
Excellent suggestion! We will recast our case studies section as the Results section, and include a new section 4 - Discussion.

Line 185: "Fatalities ... from the earthquake itself (Budimir et al., 2014" ! "Fatalities from co seismic mass wasting can be up to an order of magnitude greater than fatalities resulting from the earthquake itself (Budimir et al., 2014"

Response:
We will accept this suggestion and modify the manuscript.

Line 188: ": : : are not well understood for this event that occurred ..." !"are not well understood on account of this event occurring in a rural and remote : : :.

Response:
We will accept this suggestion and modify the manuscript.

Line 201: "HazMapper provides : : :.evaluation of empirical relationships between parameters such as moment magnitude ..." Seems to me this is about future work and so is out of the scope of the case studies presented here which function as a kind of results section of the paper. Again move to a "potential future work" section in the discussions or conclusions.

Response:
We will move this section to a paragraph in the Discussion or Conclusions section.
Paragraph starting 205: again mixing up future work and case studies. This part feels a bit like a research proposal.

Response:
We will move this section to a paragraph in the Discussion or Conclusions section.

Line 224: Again, referencing future potential – these aren't really results of the case studies and should be moved to the discussion.

Response:
We will move this section to a paragraph in the Discussion or Conclusions section.

Line 243: Future potential again

Response:
We will move this section to a section in the Discussion or Conclusions section.

Line 290: "advancement" ! "advance", "met with obscurity" ! "hindered

Response:
We will accept this suggestion and modify the manuscript.

Line 306: "democratize" see previous comments

Response:
"Data democratization" is a common term in the data science and technology sectors. This is the idea of making data available and accessible to everyone. As such, we respectfully disagree with this comment and prefer the current language.

Line 312: "...suggesting the extents..." this seems to be the core of what the approach presented here does, and it needs to be highlighted more

Response:
It is not clear to us what the reviewer means by this comment. We added a sentence to clarify that HazMapper is not a semi-automated method.

Line 314. " rapid response ..." it seems to me the appropriate timescale for something to be considered a "rapid response" for a natural hazard, at least as far as nonacademics are going to be concerned, is the timescale on which people's lives are lost as a result of the event and its aftermath, and two weeks is actually quite a long time in this context?

Response:

In mass wasting events, lives and property are lost within minutes, and even in resource-rich nations, cleanup efforts take days to weeks. In this context, we hold that identifying likely debris flows from a single large event in a developing nation within 2 weeks does qualify as rapid response.

Line 316: verification of maps produced by HazMapper could form part of a discussion section?

Response:
Verification is premature for a multispectral satellite index map. Future iterations of HazMapper that include semi-automation will require a full accuracy assessment.

Line 355: "See letter to the editors" - is this appropriate to have under "code and data availability", and will the letter be available to readers when the manuscript is finally published? Should any important information in the letter be moved into this section?

Response:
This comment was for the peer-review process to ensure the reviewers had a chance to gain access to HazMapper. It was never intended for publication and will be removed from the revised manuscript.

Equations 1 and 2 – do equations in this journal have captions like figures? If not, I would replace the captions with text in the main article, something like "where x is the variable for blah, y is the rate of blablah, and z is the magnitude of blahblahblah".

Response:
Excellent suggestion - we will revise the manuscript to this effect.

Figure 1. "Heads up digitize" - again, is this different from regular digitization?

Response:
Yes, it is different from manual digitizing (http://wiki.gis.com/wiki/index.php/Digitizing), however, we concur that in the modern computational environment, heads up digitizing is becoming the norm. To this end, we will accept this suggestion and modify the manuscript.

Figure 3. "All maps have same orientation" - unnecessary, delete – add N arrows to figs b,c,d if worried about ambiguity.

Response:
We prefer to not obscure the figures with unnecessary north arrows but still indicate to the reader that all figures have the same orientation. We would like to leave this as-is.

I was unable to reproduce the maps in this figure in HazMapper– it seems as if perhaps the rdNDVI is being thresholded somehow and overlain on the greenest pixel images in ArcMap? If this is the case, it is a bit misleading to be presenting this as an example of the successful use of HazMapper. Presumably it would be possible to add such a facility to HazMapper with relatively little effort, otherwise the figures should really be replaced with screenshots from HazMapper, (or at the very least it needs to be made more explicit that this is a more "advanced" analysis performed for validation purposes only, and that the facility will not be available to non-expert users within the HazMapper interface itself).

Response:
We have tested some user-driven color ramps in HazMapper and will implement in the future, but for the first iteration, believe a generic -50 to +50% change in rdNDVI red to blue color ramp best suits most users. This color ramp saturates at -50 and +50%. Values greater than this will have the same color at a 50% decrease or gain in rdNDVI. The download functionality of HazMapper is intended for users to download data for use in other geographic information systems software platforms (e.g., ArcGIS, QGIS, GRASS GIS, etc.,). We have posted our ArcGIS Pro compatible symbology files used to render the data presented in the figures on our website: https://hazmapper.org/resources-faq/

Additionally, a new figure 8 to address a comment from RC1 will include the default layout from HazMapper and a draft of this figure is included below.

[Figure]

**Figure 8.** $rdNDVI$ change detection images and greenest pixel composites following 26 December 2004 tsunami in Indonesia resulting from the Sumatra–Andaman earthquake. Parameters - Dataset: Landsat-7, Event Date: 26 December 2004, maximum cloud cover: 100%, and slope threshold: 0.01°. Pre-Windows: 1, 2, and 12 months, Post-Window: 1, 2, and 12 months for panels A, B, and C, respectively. $rdNDVI$ across the event illustrates tsunami inundation zone and resulting loss in vegetation. Striping in the data from the scan-line corrector failure in Landsat-7 is evident in a short look window (e.g. 1 month, A), but these artifacts are reduced by increasing the pre- and post-event windows (B and C). By 12-month pre-post periods (C), the striping is significantly reduced in results, however, vegetative recovery is also present in this longer post-event cycle that captures the first growing season following the tsunami. Screenshots of HazMapper application, example accessible at https://go.ncsu.edu/hazmapper-indonesia-tsunami.

---

## Author Response (AR1)

**Author's Response:**

NHESS Editors and Reviewers,

Thank you for the reviews of this manuscript and helpful comments. We have made significant revisions to the manuscript consistent with your comments and feel this has improved our initial submission.

- Many grammatical changes have been made, as highlighted in the marked-up version below.
- The most considerable change comes by way of including a discussion section. Much of this material was originally in the former Case Studies section. It now provides a more explicit discussion of the benefits and limitations of HazMapper. We have also clearly addressed some limitations of HazMapper identified by the reviewers, such as the lack of utility in non-vegetated environments.
- We have removed the Earth Engine vs. Traditional GIS Environments section because, after reflection, we agree with the reviewers; this section did not provide much strength to the manuscript.
- We have acquired a non-commercial, research-only license for the source code and have opened the code and app to the public. We encourage you to explore the application and read through the source code. Both are linked in the manuscript and provided here for ease of reference.
  - Application: https://cmscheip.users.earthengine.app/view/hazmapper
  - Source code:https://doi.org/10.5281/zenodo.4103348
- We have made minor changes to the figures and have added one new one (Figure 8) to help illustrate how our methods deal with the analysis of Landsat-7 data, which experienced the scan-line corrector failure.

Below, please find the marked-up version of the manuscript where blues indicate additions, and reds indicate removals from the original manuscript file. Behind this marked-up version are our previous point-by-point responses to comments. We look forward to hearing your comments on this revised manuscript.

Sincerely,

Corey Scheip and Karl Wegmann

**HazMapper: A global open-source natural hazard mapping application in Google Earth Engine**

Corey M. Scheip1,2 and Karl W. Wegmann1,3

1Department of Marine, Earth, and Atmospheric Sciences, North Carolina State University, Raleigh, NC, 27695, USA
 2North Carolina Geological Survey, Swannanoa NC, 28778, USA
 3Center for Geospatial Analytics, North Carolina State University, Raleigh, NC, 27695, USA

Correspondence: Corey M. Scheip (cmscheip@ncsu.edu)

[revised manuscript text omitted]

- 50 processing, and visualization (Gorelick et al., 2017). Aside from the obvious advantages of utilizing Google's computational resources, a key component of Google Earth Engine is the data catalog, removing the requirement for users to download and maintain large datasets, which are often gigabytes to petabytes in size. Google Earth Engine allows users to create public-facing applications, further increasing the accessibility of processing routines to specialists in the field as well as the scientific-curious public.
- 55 HazMapper (Hazard Mapper) is an open-access application developed in Google Earth Engine for the rapid characterization of natural disasters tailored to both the scientific and emergency management communities (Figure 1). HazMapper is

useful for monitoring landscape change that results in the disruption of surface vegetation . removal or recovery of terrestrial vegetation associated with a natural disaster or human activities. The platform is not currently suitable for use in non-vegetated environments (e.g. polar, high altitude, or desert regions).

- 60 While the underlying mathematics are not entirely novel, HazMapper democratizes multi-spectral satellite data processing with an emphasis on locating and characterizing natural hazards in vegetated regionsfor the evaluation of natural hazards by leveraging the accessibility and computational power of Google Earth Engine. Select case studies are discussed here and include rainfall triggered both rainfall and seismically-triggered mass wasting, seismically triggered mass wasting, wildfireburn extents, pyroclastic flows, and lava flow inundation following a volcanic wildfire, pyroclastic flow, and landscape burial by lava
- 65 flows resulting from a fissure eruption. HazMapper is intended to foster community development surrounding rapid natural hazard mapping and characterization and we invite new ideas on leveraging the platform to easily, rapidly, and more precisely monitor Earth's dynamic surface and associated natural hazards. HazMapper is In the aftermath of large natural disasters, the level of emergency response can vary based on available resources for the region or country. For example, response efforts for a large landslide disaster in the United States, (e.g. 2014 Oso landslide, WA or 2019 Montecito, CA debris flows) can garner
- 70 the attention and resources of local, state, and federal government agencies (Scholl and Carnes, 2017). In less affluent regions, however, response efforts may be less intensive and timely, risking increased loss of life. An overarching goal of HazMapper is to bring modern, rapid scientific analysis and computing power to regions with less adequate resources. HazMapper is publicly accessible at https://hazmapper.org/.

**2 Design Principles**

[revised manuscript text omitted]

The results of the processing routine indicate a normalized percentage of NDVI gained or lost. The normalization parameter
follows Ambrose et al. (2019)includes NDVIppre to account for pixels that had a low NDVI value before the event (e.g. existing mass wasting scars, urban areas). The metric follows Norman and Christie (2020), who propose this method for addressing pixel fractionation and fractional pixels, where the NDVI signal response of vegetation is not consistent across a single pixel (e.g. both forest and grasses), and the non-linear responsiveness of NDVI. Results may exceed +/- 100%. Loss-rdNDVI results of < -100% is possible due to the ability of VIR to increase to greater than the NIR value, causing a polarity change</li>
of NDVI. Results illustrate areas of the landscape that have either gained (increase in NDVI pixel-values) or lost (decrease in NDVI pixel-value) vegetation across the event as constrained by the pre-and-post event window date ranges. For visualization purposes, HazMapper applies a color-scale within the domain of -50% to +50%, simplifying the analysis to highlight areas with significant-vegetative loss or gain. Additionally, an inspector tool allows a user to click anywhere within the map domain, upon which a latitude-longitude coordinate pair and the rdNDVI pixel value will be returned interactively to the user.

135
$$rdNDVI = \left(\frac{NDVI_{post} - NDVI_{pre}}{\sqrt{NDVI_{pre} + NDVI_{post}}}\right) \times 100$$

140

Equation 2. Relative-difference in NDVI (rdNDVI) equation. NDVIpre = NDVI image of the pre-event greenest pixel composite. NDVIpost = NDVI image of the post-event greenest pixel composite. The three resulting data layers (greenest pixel composite from pre- and post-event, and rdNDVI) and a shuttle radar topography mission (SRTM) derived 30-m resolution hillshade layer (for areas between 60°N and 60°S latitude) are added to the standard Google Earth Engine layer pane. These layers can be toggled on/off and the transparency increased or decreased their transparency modified with a slider to help with visualization. Available basemaps on the platform include Google's default suite of road, terrain, and satellite maps.

Heads-up digitization Digitization of areas of interest from the map domain, for example, debris flow initiation sites, landslide extents, or potential wildfire burn areas, can be recorded using Google's default mapping tools. Points, lines, and polygons may be digitized in one or multiple layers. During download from HazMapper, these digitized geometries can be saved as a

145 keyhole-markup language (KML) file for viewing in Google Earth or sharing amongst an emergency response team or between research colleagues.

HazMapper includes an example panel in the lower left of the tool, pointing the user to five real-world natural hazard case studies (Figure 1). The panel is intended to serve as a learning platform for new users to work with curated examples to explore these events.

150 Additional sharing of HazMapper results is made available via the use of variable-tags within the URL. During usage of the tool, URL tags for the required event parameters are updated. Sharing of the updated link with a colleague or research partner

allows that person to open HazMapper to the same viewport and updates the map function to the same event parameters. For this article and shared finds at HazMapper.orgthe case studies discussed here, we have utilized the North Carolina State University Go Links URL service. For example, https://go.ncsu.edu/hazmapper-kenya directs the user to the curated example

155 of rainfall-triggered debris flows located in west Pokot County, Kenya, in November 2019 (see Section 4.1.1). All case studies discussed in this paper are available pre-loaded from the HazMapper launch screen, under the "Show Examples" tab, as well as at or as direct URL links (see Code and Data Availability.

Data download is an important component of HazMapper. This function allows for enhanced use of the further analysis of processing results, including incorporation into emergency operation mapping platforms and advanced scientific analysis

160 . Downloads are distributed as single zip files containing single-band georeferenced TIF files. or visualization. The user can download the 1) rdNDVI image, 2) pre-event and 3) post-event greenest-pixel composite images, 4) elevation and hillshade images derived from the global 30-m SRTM dataset and/or 5) any user-digitized geometries delineating points or areas of interest. Raster data layers are distributed as geographic tagged image file format (GeoTIFF) and digitized geometries as KML.

**165 3 Earth Engine vs. Traditional GIS Environments**

In the aftermath of large natural disasters, the level of emergency response can vary based on available resources for the region or country. For example, response efforts for a large landslide disaster in the United States, (e.g. 2014 Oso landslide, WA or 2019 Montecito, CA debris flows) can garner the attention and resources of local, state, and federal government agencies (Scholl and Carnes, 2017). In less affluent regions, however, response efforts may be significantly less intensive and timely, risking increased loss of life. An overarching goal of HazMapper is to bring modern, rapid scientific analysis and computing

170

power to regions with less adequate resources.

Inventorying impacts following a natural disaster can be incredibly useful for researchers. Inventory efforts are time and resource-intensive (e.g., purchasing high-resolution aerial imagery or performing significant field verification efforts) and often experience time delays following the disaster (Malamud et al., 2004). HazMapper provides an opportunity for researchers to

- 175 perform preliminary inventory work without field visits and without paying for commercial data products. Additionally, it allows researchers to time-stamp ground conditions. This is useful to understand the progression of disasters, such as continued wildfire spread or mass wasting initiated by earthquake aftershoeks. Traditional GIS analysis requires significant digital storage capacity, computing power, and training. Even with modern personal computers, processing can take hours to days. Individual scenes of Sentinel-2 datasets, for example, are typically hundreds of megabytes. Processing of these data also result in a
- 180 fixed-extent output. Google Earth Engine's architecture initiates geospatial processing updates as the user navigates the map and, as such, HazMapper can be used to quickly evaluate spatially expansive hazards. HazMapper's source code initiates data processing on remote servers without requiring any specialized or licensed software and can be performed on any computer with an internet-browser and internet-connection. No software downloads are required and typical processing times on HazMapper

are less than 1-2 minutes, allowing responders or researchers to adjust variables and visualization parameters to rapidly assess

185 the potential impact of natural hazards at a specific location or across a region of interest.

**3 Results – Case Study Examples**

For many decades, researchers have utilized satellite-observed vegetation losses as a proxy for change on Earth's surface. The usefulness of satellite-observations in the scientific community are difficult to overstate and future developments (e.g. Hoffman et al., 2016) are increasing the need for enhanced processing methodologies and techniques. Some traditional

[revised manuscript text omitted]

- 225 of debris flows, rdNDVI also captured assumed hyper-concentrated streamflow, where riparian vegetation was removed from riparian vegetation loss and sedimentation along the banks of lower-gradient rivers as they drain the mountainous terrain where the mass wasting occurred. Agricultural harvesting and planting activities are also apparent in the rdNDVI results, evident by their position on low-relief terrain outside of the drainage channels (Figure 3D).
- Debris flows in western Kenya are not unprecedented and the Elgeyo Escarpment, the steep terrain along the western edge of the Great Rift Valley, has experienced significant historical landsliding (Maina-Gichaba et al., 2013). The steep slopes result from normal faulting associated with extension of the Great Rift Valley and are overlain by up to 3 km of Miocene sediments and lava flows that in the tropical climate have weathered to produce thick, residual soils (Maina-Gichaba et al., 2013). Average rainfall for the area is >800mm annually (Maina-Gichaba et al., 2013), but is likely greater along steep valley walls due to orographic lifting (Hession and Moore, 2011). The resulting thick soils and ample moisture have led to significant agricultural
- 235 development and the region is heavily dissected by a patchwork of agricultural fields. Occasionally, heavy rainfall couples with the thick sediment packages to produce shallow, primarily rainfall-triggered, mass wasting that initiates primarily high on valley walls.

The West Pokot County debris flow HazMapper example illustrates the pace at which rapid repeat-cycle optical imagery can be utilized to aid in the identification of hazard-stricken areas. Figure 3 demonstrates the results after a 0.5-month post-event window. The timing of suitable datasets for an initial look at disaster impacts will depend on the timing of a disaster relative to acquisition schedules of the Sentinel-2 or Landsat platforms and atmospheric conditions (e.g. cloud cover during or following a rainfall-triggered mass wasting event). Acquisition schedules are publicly available for both platforms and can be used in

**3.1.2 Seismically-triggered mass wasting, Southern Highlands, Papua New Guinea, Mw 7.5, 25 February 2018**

conjunction with HazMapper to help responders understand when suitable datasets may become available.

245 On 25 February, 2018, a  $M_w$  7.5 earthquake struck in the Southern Highlands of Papua New Guinea (PNG) along the Papuan Fold and Thrust Belt (Wang et al., 2019), triggering thousands of mass wasting events, damming the Tagari River, and impacting numerous communities across the region. Over a span of 2 months, 5 aftershocks of  $M_w$  >6 struck the same region (Wang et al., 2019). Two Nearly three years after the event, a published mass wasting inventory is not availablewhile communities are still impacted from the sequence of earthquakes and hillslope failures . Fatalities from coseismic mass wasting events can

- 250 increase significantly, can be up to an order of magnitude, greater than fatalities resulting from the earthquake itself (Budimir et al., 2014). The 2018 Papua New Guinea earthquake and associated mass wasting resulted in at least 160 fatalities (Wang et al., 2019), but the individual contributions (e.g. building collapse, burial by hillslope mass movements, etc) are not well understood for this event that occurred on account of this event occurring in a rural and remote part of the country.
- Seismic shaking is a primary triggering mechanism for mass movement mobilization on steep mountain terrain. Coseismic mass wasting, therefore, strongly influences erosional budgets of mountain belts (Hovius et al., 1997; Keefer, 1994; Korup et al., 2010; Hilton et al., 2008). Keefer (2002) has demonstrated an empirical relationship based upon a global dataset between the moment magnitude of a mainshock and the maximum distance from the epicenter that seismically-induced landslides are likely to be observed for the entire earthquake sequence (including aftershocks). For this Mw 7.5 earthquake, the corresponding predicted maximum distance is approximately 300 km. HazMapper was utilized to rapidly assess regions within several tens
- of kilometers from the epicenter and hundreds of slides and flows were located (Figure 4). Additional mass wasting was noted when expanding the analysis window to the predicted 300 km maximum distance based on the earthquake magnitude. Furthermore, we noted possible coseismic slides and flows as far as several hundred km west of the epicenter in the Maoke Mountains of Indonesia. Mass wasting is common in the region and these events could have unique triggers, however, (e.g. rainfall triggered). However, restricting pre- and post-event time windows to as little as 2 months bracketing the Mw 7.5
- 265 mainshock demonstrates consistent timing with the 25 February, 2018 earthquake. HazMapper provides both a past- and futurelooking approach to develop robust spatial and temporal catalogs of coseismic mass wasting and the evaluation of empirical relationships between parameters such as moment magnitude, modified Mercalli intensity scale, depth, focal mechanism, regional lithology, topographic position, land use type, etc. as functions of the distance from an epicenter.
- Due to difficulties in ascertaining high-temporal-resolution sequences of mass wasting events following seismic shaking, it is typically difficult to determine if particular events were triggered by just the mainshock or also by aftershocks. Thus, research to date has focused on earthquake sequences, inclusive of all associated shaking (e.g. Keefer, 2002). HazMapper allows researchers to temporally constrain landscape change and in certain circumstances, may be useful for understanding hillslope failure sequences when large aftershocks follow the main event for large earthquakes that may include landslide inducing aftershocks . Future research should consider utilizing these time-stamped change detection images to understand the progression of failures during an earthquake sequence.
  - While identifying vegetative vegetation loss for locating geohazards is a key characteristic of a mass wasting event response, identifying subsequent vegetation recovery can serve as a proxy for the reduction of associated hazard (Shen et al., 2020). Simple modifications to event parameters in HazMapper, for example by changing the "event date" to a time after the occurrence of the disturbance event, can aid in observing vegetative vegetation recovery in landslide scars, suggesting
- establishment and growth of early successional species like grasses and shrubs (Figure 4-D). These stabilizing root masses buttress further soil loss and erosion, and, thus, decrease the associated downslope sediment transport from the zone of mass wasting.

**3.2 Wildfire**

[revised manuscript text omitted]

**345 3.3.2 Effusive lava flows — Lower East Rift Zone (LERZ)eruption , Kīlauea Volcano, Hawaii, USA, May-September 2018**

Kīlauea is a basaltic shield volcano built from lavas derived from deep mantle driven processes. The magma feeding the volcano is distributed through a network of shallow rift structures and was pooled in a lava lake at its summit until commencement of the 2018 eruptive sequence. Eruptive characteristics have varied through time including a combination of periods of summit and/or
rift eruptions, and caldera collapse, in-fill, and overflow (Holcomb, 1987). The most recent 2018 caldera collapse-rift eruption sequence was well captured by a dense array of scientific instrumentation and social networking, adding significant information to our present understanding of the Kīlauea complex (Neal et al., 2019). The 2018 event culminated in the inundation of 35.5 km2 of Hawaii's Big Island and the destruction of hundreds of homes. Fortunately, there are no known fatalities from the event, likely due to the slow moving nature of the eruption and the significant resources applied during the disaster management response efforts.

HazMapper was utilized to observe surface changes within the Lower East Rift Zone (LERZ) following the cessation of the rift flank eruption sequence (Figure 7). Utilizing 30-meter Landsat data, the observed vegetation loss extending east and southeast from the LERZ approximates the published flow field from the 2018 eruption (Hawaiian Volcano Observatory staff, 2018). Efforts to utilize HazMapper to monitor the advancement advance of the lava flows were met with obscurity hindered

- 360 due to persistent cloud cover and volcanic gas emissions during the eruption. Additionally, the east-southeast extents of the lava flows generated additional landmass off of the coast of Hawaii, but with no vegetation to lose, this landscape change was not detected using HazMapper. Future code modifications may allow for the identification of the additional landmasses added to the island of Hawaii following the eruptions.
- The 2018 Kīlauea eruption response benefited from significant resource application by way of the existing Hawaii Volcano Observatory and the associated resources of the U.S. federal and Hawaii state governments and associated scientific and resource protection agencies. This example, therefore, is highlighted to perform a first-order comparison of the kind of results available with HazMapper, a free and open-access toolset to an on-the-ground effort with significant financial, personnel, equipment, and computing resources and attention (Figure 7) . For eruptions with less global attention or in more remote regions, remote sensing results like those available with HazMapper alone may approximate lava flow inundation extents,
- 370 guiding future response efforts or scientific research around the event. Furthermore, the utilization of a consistent analysis platform between many eruptions may aid in volcanic research globally.

HazMapper is

**4 Discussion**

**4.1 Limitations**

[revised manuscript text omitted]

Future development of HazMapper will leverage new datasets data sets as they become available. The initial release 485 includes options to analyze Sentinel-2, Landsat 8, and Landsat 7 datasets data sets . HazMapper's underlying codeset source code is designed to be easy to add forthcoming datasets easily incorporate multi-spectral imagery data from forthcoming missions , such as Landsat 9 that is anticipated to launch in December 2020 (McCorkel et al., 2018).

At the time of this publication, the features discussed herein have been tested, however, it should be noted that GEE is a rapidly evolving technology. As changes are made within GEE, we will maintain HazMapperto the best of our ability, including maintaining existing functionality and adding functionality as technology permits. HazMapper

**5 Conclusions**

[revised manuscript text omitted]

---

## Author Response (AR2)

**Author's Response:**

NHESS Editors and Reviewers,

Thank you for the reviews of this manuscript and helpful comments. We have made minor revisions to the manuscript consistent with your comments and feel this has improved our revised submission. There have been no additional changes to the tables, figures, or supplement.

Below, please find our point-by-point response to reviewer comments #3 and #4. We look forward to hearing your comments on this revised manuscript.

Sincerely,

Corey Scheip and Karl Wegmann

**Response to anonymous reviewer / RC3 / Referee #3**

We sincerely thank the anonymous reviewer for thoughtful feedback on this manuscript. We especially appreciate these reviews considering this paper presents the first iteration of a new research tool.

Color legend
RC3 - black
Author response - blue

**RC3 Comments**
The manuscript describes HazMapper, a Google Earth Engine based application to visualize natural hazards based on vegetation change pre and post event. I believe this system will be very useful for preliminary analysis specially for non-scientists to get an understanding and identifying areas affected by natural hazards. Although validation and accuracy metrics are not available in present form, this tool is invaluable for situational awareness. This manuscript has already gone through first round of reviews. I only have minor comments.

Response:
Thank you for your overall assessment of this platform. We agree in this early-stage, the focus is situational awareness and look forward to advancing the platform in subsequent iterations with an eye toward quantitative assessments.

Line 10:
Authors states this tool is useful also for historical natural disasters. While true, historical events will benefit form products which can be used for scientific analysis and modeling. This iteration of Hazmapper has to rely on hand mapping within the application. Doable for a small event but highly cumbersome if needed to be done over larger areas. I think it is more suitable towards visualizing events which will occur in future.

Response:
Thank you for this comment. We have added "and visualization" to this sentence. Hand mapping is useful but can be time consuming as you point out. To alleviate this, the supplement to this paper provides an example of thresholding rdNDVI data (downloaded from HazMapper) to quickly assess for vegetation loss over large areas.

Line 55 – 60:

Justification for developing HazMapper based on level of emergency response and available resource between affluent and non-affluent regions is not correct. I urge authors to revisit this section and come up with a better motivation for developing HazMapper.

Response:
Thank you for this comment. Our motivations have not changed, but we have modified this sentence to read as:
*An overarching goal of HazMapper is to leverage rapid scientific analysis and computing tools for global natural hazards awareness.*

Equation 1:
Why use VIR instead of Red which is more common?

Response:
Thank for you for this comment. We have replaced "VIR" in the equation with "Red" and updated the variable definitions in the following line. From the updated manuscript:

$$NDVI = \left( \frac{NIR - Red}{NIR + Red} \right) \tag{1}$$

where NIR is the near-infrared response and Red is the visible red response.

Line 163:
How does one judge an image to be suitable?

Response:
Thank you for this comment. For these optical methods, a key assessment of image suitability is cloud-cover. We have removed the word "suitable" and simple stated "Sentinel-2 images with limited cloud cover."

Line 290:
How is data latency for Sentinel and Landsat in GEE. Does this add to the time mentioned here? This should be discussed.

Response:
Excellent point - we have added a sentence to this effect:
*Imagery is typically available on GEE within approximately 24-hours of its collection by the satellite*

Line 288:

Do authors envision machine learning based HazMapper to semi automate mapping of every natural hazard described here. Visualization as it is now is not an issue but when it comes to automation, different natural hazards will require different setup. Some clarification on this will be useful.

Response:
Thank you for this comment. Our machine learning program will first focus on semi-automated landslide detection. We have added "for landslide identification" to the manuscript to clarify this.

**Response to anonymous reviewer / RC4 / Referee #2**

We sincerely thank the anonymous reviewer for thoughtful feedback on the revised manuscript.

Color legend
RC4 - black
Author response - blue

**RC4 Comments**
I thank the authors for their corrections, the paper is greatly improved, issues with the structure and style of the paper are resolved, and I defer to them on issues of terminology. However I feel my 2nd topical comment, that I don't think the website succeeds in making these techniques accessible to non expert users (i.e. democratizing them, which I understand now is the correct term and the point of the paper). This could be remedied with a basic walk through mode, for which there are a range of libraries and which can be trivially added with minimal effort (a day or two at the most). Even a simple page of text describing them, accessible from a link (as I think you are suggesting with http://hazmapper.org/learn) would do, which would be even simpler. So I recommend minor revisions to resolve this one outstanding point.

Response:
Thank you for reviewing our revised manuscript and suggesting the build-out of our http://hazmapper.org/learn portion of our website. Following your comment, we have posted text, images, and GIFs to walk a new user through our rainfall-triggered debris flow example from Kenya. We plan to continue the development of training, teaching, and outreach materials following publication. We did not make any changes to the manuscript resulting from this comment.

---

## Author Response (AR3)

**Author's Response:**

NHESS Editors and Reviewers,

Thank you all for your time and energy in reviewing this manuscript. Your comments have greatly improved the original version of this manuscript and we cannot thank you enough! We look forward to continuing to work on this project.

Sincerely,

Corey Scheip and Karl Wegmann

**Response to the Editor**

Dr. Parks,
Thank you for your review of this manuscript and the minor suggestions.

Color legend
Editor - black
Author response - blue

**Editor Comments**
line 231 "As such, volcanologists have been using remote-sensing tools, particularly multi-spectral satellite data, as early as the mid-1980's to monitor volcanic heat signatures as precursors to eruptive activity (Rothery et al., 1988)." Please reword slightly.

The primary application of remote-sensing now used by volcanologists is mapping ground deformation related to magmatic intrusions to determine the parameters of the intrusion - location, geometry and volume change - and to map magma migration prior to the onset of an eruption. It is also routinely used for mapping co-eruptive deformation and estimating magma flow rates.

Response:
We have revised this section of the manuscript to include the use of SAR data for ground deformation monitoring

*Volcanologists began using remote-sensing tools, particularly multi-spectral satellite data, as early as the mid-1980's to monitor volcanic heat signatures as precursors to eruptive activity (Rothery et al., 1988). More recently, synthetic aperture radar data aid in monitoring ground deformation associated with magmatic intrusions or eruptions (Hooper et al., 2004).*

line 385 "possess"

Response:
Thank you for this catch! We have corrected this spelling error.